# Bayesian Metric Learning for Uncertainty Quantification in Image Retrieval

**Frederik Warburg**[*]
Technical University of Denmark
frwa@dtu.dk

**Marco Miani**[*]
Technical University of Denmark
mmia@dtu.dk

**Silas Brack**
Technical University of Denmark
silasbrack@gmail.com

**Søren Hauberg**
Technical University of Denmark
sohau@dtu.dk

## Abstract

We propose a Bayesian encoder for metric learning. Rather than relying on neural amortization as done in prior works, we learn a distribution over the network weights with the Laplace Approximation. We first prove that the contrastive loss is a negative log-likelihood on the spherical space. We propose three methods that ensure a positive definite covariance matrix. Lastly, we present a novel decomposition of the Generalized Gauss-Newton approximation. Empirically, we show that our Laplacian Metric Learner (LAM) yields well-calibrated uncertainties, reliably detects out-of-distribution examples, and has state-of-the-art predictive performance.

## 1 Introduction

Metric learning seeks data representations where similar observations are near and dissimilar ones are far. This elegantly allows for building retrieval systems with simple nearest-neighbor search. Such systems easily cope with a large number of classes, and new classes can organically be added without retraining. While these retrieval systems show impressive performance, they quickly, and with no raised alarms, deteriorate with out-of-distribution data [38]. In particular, in safety-critical applications, the lack of uncertainty estimation is a concern as retrieval errors may propagate unnoticed through the system, resulting in erroneous and possibly dangerous decisions.

We present the Laplacian Metric Learner (LAM) to estimate reliable uncertainties of image embeddings as demonstrated in Fig. 1. We learn a distribution over the network weights (weight posterior) from

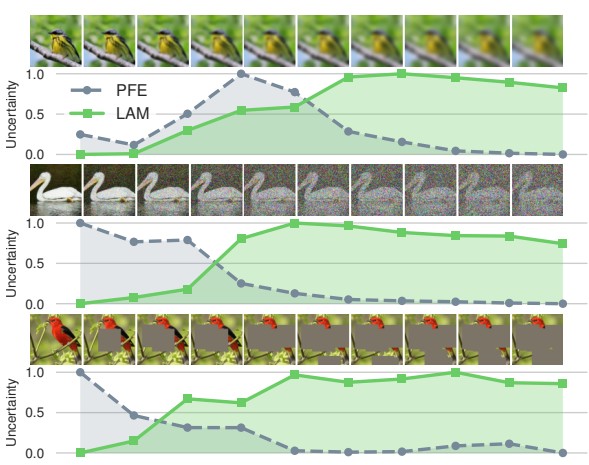

Figure 1: **Reliable stochastic embeddings.** LAM (ours) estimates reliable uncertainties of latent image embeddings that reflect the amount of blur, noise, or occlusion in the input image.

---

[*]Equal contribution

37th Conference on Neural Information Processing Systems (NeurIPS 2023).

which we obtain a stochastic representation by embedding an image through sampled neural networks. This Bayesian formulation has multiple benefits, namely (1) robustness to out-of-distribution examples, (2) calibrated in-distribution uncertainties, and (3) a slight improvement in predictive performance.

More specifically, our method extends the Laplace Approximation [27] for metric learning. We present a probabilistic interpretation of the contrastive loss [15] which justifies that it can be interpreted as an unnormalized negative log-likelihood. We propose three solutions to ensure a positive semidefinite Hessian for the contrastive loss and present two approaches to compute the Generalized Gauss-Newton [12] approximation for $\ell_2$-normalized networks. Finally, we boost our method with the online training procedure from [29] to achieve state-of-the-art performance.

We are not the first to consider uncertainty quantification in image retrieval. Seminal works [38, 33] have addressed the lack of uncertainties in retrieval with *amortized inference* [14], where a neural network predicts a stochastic embedding. The issues with this approach are that (1) it requires strong assumptions on the distribution of the embedding, (2) the networks are often brittle and difficult to optimize, and (3) out-of-distribution detection relies on the network's capacity to extrapolate uncertainties. As neural networks extrapolate poorly [50], the resulting *predicted* uncertainties are unreliable for out-of-distribution data [10] and are thus, in practice, of limited value.

In contrast, our method does not assume any distribution on the stochastic embeddings, is simple to optimize, and does not rely on a neural network to extrapolate uncertainties. Instead, our weight posterior is derived from the curvature of the loss landscape and the uncertainties of the latent embeddings deduced (rather than learned) with sampling. Empirically, we show that this leads to reliable out-of-distribution performance and calibrated uncertainties in both controlled toy experiments and challenging real-world applications such as bird, face, and place recognition.

## 2 Related Work

**Metric learning** attempts to map data to an embedding space, where similar data are close together and dissimilar data are far apart. This is especially useful for retrieval tasks with many classes and few observations per class such as place recognition [46] and face recognition [37] or for tasks where classes are not well-defined, such as food tastes [48] or narratives in online discussions [4].

There exist many metric losses that optimize for a well-behaved embedding space. We refer to the excellent survey by [30] for an overview. We here focus on the *contrastive loss* [15]

$$\mathcal{L}_{\mathrm{con}}(\theta) = \frac{1}{2}\|f_\theta(x_a) - f_\theta(x_p)\|^2 + \frac{1}{2}\max\left(0, m - \|f_\theta(x_a) - f_\theta(x_n)\|^2\right), \qquad (1)$$

which has shown state-of-the-art performance [30] and is one of the most commonly used metric losses. Here, $f_\theta$ is a neural network parametrized by $\theta$ which maps from the observation space to the embedding space. The loss consists of two terms, one that attracts observations from the same class (*anchor* $x_a$ and *positive* $x_p$), and one that repels observations from different classes (*anchor* $x_a$ and *negative* $x_n$). The margin $m \geq 0$ ensures that negatives are repelled sufficiently far. We will later present a probabilistic extension of the contrastive loss that allows us to learn stochastic, rather than deterministic, features in the embedding space.

**Uncertainty in deep learning** is studied across many domains to mitigate fatal accidents and allow for human intervention when neural networks make erroneous predictions. Current methods can be divided into methods that apply amortized optimization to train a neural network to predict the parameters of the output distribution, and methods that do not. The amortized methods, best known from the variational autoencoder (VAE) [19, 36], seem attractive at first as they can directly estimate the output distribution (without requiring sampling), but they suffer from mode collapse and are sensitive to out-of-distribution data due to the poor extrapolation capabilities of neural networks [31, 10]. 'Bayes by Backprop' [2] learns a distribution over parameters variationally but is often deemed too brittle for practical applications. Alternatives to amortized methods includes deep ensembles [23], stochastic weight averaging (SWAG) [28], Monte-Carlo dropout [13] and Laplace Approximation (LA) [24, 27] which all approximate the generally intractable weight posterior $p(\theta|\mathcal{D})$ of a neural network. We propose to extend LA to metric learning.

**Laplace approximations (LA)** can be applied for every loss function $\mathcal{L}$ that can be interpreted as an unnormalized log-posterior by performing a second-order Taylor expansion around a chosen weight

vector $\theta^*$ such that

$$\mathcal{L}(\theta) \approx \mathcal{L}^* + (\theta - \theta^*)^\top \nabla \mathcal{L}^* + \frac{1}{2}(\theta - \theta^*)^\top \nabla^2 \mathcal{L}^*(\theta - \theta^*), \qquad (2)$$

where $\mathcal{L}^*$ is the loss evaluated in $\theta^*$. Imposing the unnormalized log-posterior to be a second-order polynomial is equivalent to assuming the posterior to be Gaussian. If $\theta^*$ is a MAP estimate, the first-order term vanishes, and the second-order term can be interpreted as a precision matrix, the inverse of the covariance. Assuming $\theta^*$ is a MAP estimate, this second-order term is negative semi-definite for common (convex) supervised losses, such as the mean-squared error and cross-entropy. Recently, Daxberger et al. [8] demonstrated that post-hoc LA is scalable and produces well-behaved uncertainties for classification and regression. The Laplacian Autoencoder (LAE) [29] improves on the post-hoc LA with an online Monte Carlo EM training procedure to learn a well-behaved posterior. It demonstrates state-of-the-art uncertainty quantification for unsupervised representation learning.

**Uncertainty in metric learning** is not new [44], but the majority of recent methods apply amortized inference to predict distributions in the embedding space [47, 3, 5, 33, 38, 39, 41], making them sensitive to mode collapse and out-of-distribution data. Alternatives like deep ensembles [42] and Monte-Carlo dropout [43] suffer from increased training time, poor empirical performance, and limited Bayesian interpretation [8]. We explore LA in metric learning and attain state-of-the-art performance.

# 3 Laplacian Metric Learning

To perform Bayesian retrieval, we estimate the weight posterior of the embedding network $f_\theta$ such that we can sample data embeddings to propagate uncertainty through the decision process. The embedding network is parametrized by $\theta \in \Theta$ and trained with the contrastive loss. The network maps an image $x \in \mathcal{X} := \mathbb{R}^{HWC}$ to an embedding $z \in \mathcal{Z}$, which is restricted to be on a $Z$-dimensional unit sphere $\mathcal{Z} := \mathcal{S}^Z$. This spherical normalization is often done in retrieval to obtain faster retrieval and a slight performance boost [1, 35]. Fig. 2 illustrate our Bayesian mapping from image to latent space.

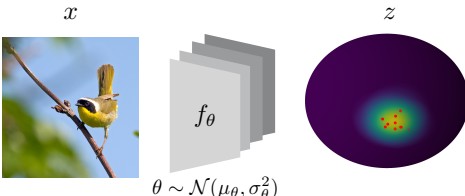

Figure 2: **Model overview**. We learn a distribution over parameters, such that we embed an image through sampled encoders $f_\theta$ to points $z_i$ (red dots) in a latent space $\mathcal{Z}$. We reduce these latent samples to a single measure of uncertainty by estimating the parameters of a von Mises-Fisher distribution.

**Laplace Approximation (LA) recap.** We apply LA to obtain the posterior over the weights $\theta$. LA comes in two flavors: (1) *The post-hoc LA is found by first training a standard deterministic network through gradient steps with the contrastive loss to find the *maximum a posteriori* (MAP) parameters $\theta^*$. Since we are in a local optimum, the first-order term in the second-order Taylor expansion (Eq. 2) vanishes, and we can define the parameter distribution as

$$p(\theta|\mathcal{D}) = \mathcal{N}\left(\theta\Big|\theta^*, \left(\nabla_\theta^2 \mathcal{L}_{\text{con}}\left(\theta^*; \mathcal{D}\right) + \sigma_{\text{prior}}^{-2}\mathbb{I}\right)^{-1}\right). \qquad (3)$$

The advantage of post-hoc LA is that the training procedure does not change, and already trained neural networks can be made Bayesian. In practice, however, we empirically observe post-hoc LA to be unstable. We refer to the large standard deviation in Table 1, Table 2, and Table 3 over random seeds. The instability stems from curvature differences between the local minimas through stochastic optimization. (2) *Online LA* [29] improves on this instability by marginalizing the LA during training with Monte Carlo EM. This helps the training recover a solution $\theta^*$ where the Hessian reflects the loss landscape. Specifically, at each step $t$ during training, we keep in memory a Gaussian distribution on the parameters $q^t(\theta) = \mathcal{N}(\theta|\theta_t, H_{\theta_t}^{-1})$. The parameters are updated through an expected gradient step

$$\theta_{t+1} = \theta_t + \lambda \mathbb{E}_{\theta \sim q^t}[\nabla_\theta \mathcal{L}_{\text{con}}(\theta; \mathcal{D})] \qquad (4)$$

and a discounted Laplace update

$$H_{\theta_{t+1}} = (1 - \alpha)H_{\theta_t} + \nabla_\theta^2 \mathcal{L}_{\text{con}}(\theta; \mathcal{D}), \qquad (5)$$

where $\alpha$ describes an exponential moving average, similar to momentum-like training. The initialization follows the isotropic prior $q^0(\theta) = \mathcal{N}(\theta|0, \sigma_{\text{prior}}^2\mathbb{I})$.

**The probabilistic contrastive likelihood.** LA expects the loss function to be a log-posterior, that is log-likelihood plus log-prior. A simple choice of the log-prior is $\|\theta\|_2^2$ (weight decay). Further, we find that the contrastive loss Eq. 1 has a probabilistic interpretation and is a negative log-likelihood on the spherical space. We define the attractive term $\mathcal{P}^{\rightarrow\leftarrow}(z|x,\theta) \sim \mathcal{N}^S(z|f_\theta(x),\kappa)$ and the repelling term $\mathcal{P}^{\leftarrow\rightarrow}(z|x,\theta) \sim \mathcal{N}^S(z|-f_\theta(x),\kappa)$ as Von Mises-Fisher distributions on the latent space spherical space, where the concentration parameter $\kappa \geq 0$. The product of these two likelihoods yields a contrastive likelihood, which is valid on the spherical space, and its negative log-likelihood is equivalent to the contrastive loss (proof and details in **??**).

The intuition, helped by electrostatics, is that the log probability density of these Von Mises-Fisher distributions (i.e. spherical Gaussians) plays the role of a potential energy. The precision parameter $\kappa$ controls the concentration of the potential energy and, consequently, the strength of the associated force. If $\kappa = 0$ the distribution is uniform, the potential is constant and the associated force is zero. On the spherical space, moreover, a repulsive term from some point is equivalent to an attractive term from the antipodal point, which in turn is mathematically equivalent (see **??**) to a negative precision $\kappa$. This is the fundamental reason behind the non positive-definiteness of the hessian.

**Hessian of the contrastive loss.** Both post-hoc and online LA require the Hessian of the contrastive loss $\nabla_\theta^2 \mathcal{L}_{\text{con}}(\theta; \mathcal{D})$. The Hessian is commonly approximated with the Generalized Gauss-Newton (GGN) approximation [12, 8, 6, 11]. The GGN decomposes the loss into $\mathcal{L} = g \circ f$, where $g$ is usually chosen as the loss function and $f$ the model function, and only $f$ is linearized [22].

However, in our case, this decomposition is non-trivial. Recall that the last layer of our network is an $\ell_2$ normalization layer, which projects embeddings onto a hyper-sphere. This normalization layer can either be viewed as part of the model $f$ (linearized normalization layer) or part of the loss $g$ (non-linearized normalization layer). The former can be interpreted as using the *Euclidean* distance and the latter as using the *Arccos* distance for the contrastive loss (see **??** and **??**). These two share the zero- and first-order terms for normalized embeddings but not the second-order derivatives due to the GGN linearization. The Euclidean interpretation leads to simpler derivatives and interpretations, and we will therefore use it for our derivations. We emphasize that the Arccos is theoretically a more accurate approximation, because the $\ell_2$-layer is not linearized, and we provide derivations in **??**.

The GGN matrix for contrastive loss with the *Euclidean* interpretation is given by

$$
\begin{aligned}
\nabla_\theta^2 \mathcal{L}_{\text{con}}(\theta; \mathcal{I}) &= \sum_{ij \in \mathcal{I}} H_\theta^{ij} = \sum_{ij \in \mathcal{I}_p} H_\theta^{ij} + \sum_{ij \in \mathcal{I}_n} H_\theta^{ij} \\
&\overset{\text{GGN}}{\approx} \sum_{ij \in \mathcal{I}_p} J_\theta^{ij\top} \underbrace{\begin{pmatrix} 1 & -1 \\ -1 & 1 \end{pmatrix}}_{:=H_p} J_\theta^{ij} + \sum_{ij \in \mathcal{I}_n} J_\theta^{ij\top} \underbrace{\begin{pmatrix} -1 & 1 \\ 1 & -1 \end{pmatrix}}_{:=H_n} J_\theta^{ij},
\end{aligned}
\tag{6}
$$

where $J_\theta^{ij} = \left( J_\theta f_\theta(x_i)^\top, J_\theta f_\theta(x_j)^\top \right)^\top$, with $J_\theta$ is the Jacobian wrt. the parameters and $H_p$ and $H_n$ are the Hessians of the contrastive loss wrt. the model output for positive and negative pairs. The first sum runs over positive pairs and the second sum runs over negative pairs *within the margin*. Negative pairs outside the margin do not contribute to the Hessian, and can therefore be ignored to reduce the computational load (**??**).

The eigenvalues of the Hessian wrt. to the output are $(0, 2)$ and $(-2, 0)$ for the positive $H_p$ and negative $H_n$ terms, so we are not guaranteed to have a positive semidefinite Hessian, $H_\theta$. To avoid covariances with negative eigenvalues, we propose three solutions to ensure a positive semidefinite Hessian. Proofs are in **??**.

**Ensuring positive definiteness of the covariance matrix**. We do not want to be restricted in the choice of the prior except to have non-zero precision, so we must ensure that $\nabla_\theta^2 \mathcal{L}_{\text{con}}(\theta^*; \mathcal{D})$ is positive semidefinite. Differently from the standard convex losses, this is not ensured by the GGN approximation [18]. Our main insight is that we can ensure a positive semidefinite Hessian $H_\theta$ by only manipulating the Hessians $H_p$ and $H_n$ in Eq. 6.

*1. Positive: The repelling term is ignored, such that only positive pairs contribute to the Hessian.*

$$
H_p = \begin{pmatrix} 1 & -1 \\ -1 & 1 \end{pmatrix}, \qquad\qquad H_n = \begin{pmatrix} 0 & 0 \\ 0 & 0 \end{pmatrix}
\tag{7}
$$

*2. Fixed: The cross derivatives are ignored.*

$$H_p = \begin{pmatrix} 1 & 0 \\ 0 & 1 \end{pmatrix}, \qquad\qquad H_n = \begin{pmatrix} -1 & 0 \\ 0 & -1 \end{pmatrix} \tag{8}$$

*3. Full: Positive semidefiniteness is ensured with ReLU,* $\max(0, \nabla_\theta^2 \mathcal{L}_{con}(\theta; \mathcal{D}))$, *on the Hessian of the loss wrt. the parameters.*

$$H_p = \begin{pmatrix} 1 & -1 \\ -1 & 1 \end{pmatrix}, \qquad\qquad H_n = \begin{pmatrix} -1 & 1 \\ 1 & -1 \end{pmatrix} \tag{9}$$

The *positive* approximation is inspired by [38], which only uses positive pairs to train an uncertainty module. The gradient arrows in Fig. 3a illustrate that negative pairs are neglected when computing the Hessian of the contrastive loss. The *fixed* approximation considers one data point at a time, assuming the other one is fixed. Thus, given a pair of data points, this can be interpreted as first moving one data point, and then the second (rather than both at the same time). We formalize this in **??**. Fig. 3b illustrate this idea when all points except $a$ are fixed. Lastly, we propose the *full* Hessian of the contrastive loss (Fig. 3c) and ensure positive semidefiniteness by computing the ReLU of the diagonal Hessian. This approximation can equivalently be interpreted as a projection into the space of psd matrixes. In practice, the Hessian scales quadratically in memory wrt. the number of parameters. To mitigate this, we approximate this Hessian by its diagonal and only apply the LA on the last layer [25, 9]. We experimentally find that the *fixed* approximation yields the best performance (Section 4).

**Hard negative mining.** Most pairs, namely the negatives outside the margin, do not contribute to the Hessian, so it is wasteful to compute their Hessian. Therefore, we use hard negative mining [30] to only compute the Hessian of pairs that have non-zero Hessian, *i.e.* the negative sample lie within the margin (illustrated with the dotted line in Fig. 3).

**Von Mises-Fisher distribution.** To obtain a single measure of uncertainty from our sampled image embeddings (red dots in Fig. 4), we fit a von Mises-Fisher distribu-

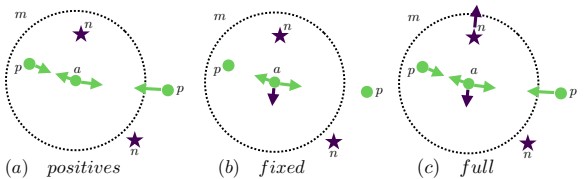

(a) *positives*     (b) *fixed*     (c) *full*

Figure 3: **Hessian approximations.** To ensure a positive semidefinite Hessian approximation we propose three approximations. In (a) only the positives $p$ contribute to the Hessian as the negatives $n$ are ignored. In (b) we consider one point at a time, e.g., only the anchor $a$ contributes. In (c) we consider all interactions.

tion. The von Mises-Fisher distribution describes a normal distribution where all probability mass lies on a $Z$-dimensional hyper-sphere. It is parametrized with a directional mean $\mu$ and a scalar concentration parameter $\kappa$, which can be interpreted as the inverse of an isotropic covariance $\kappa = 1/\sigma^2$, *i.e.*, small $\kappa$ means high uncertainty and large $\kappa$ means low uncertainty. There exist several methods to estimate $\kappa$. We opt for the simplest and most computationally efficient [40] (see **??**).

## 4 Experiments

We benchmark our method against strong probabilistic retrieval models. Probabilistic Face Embeddings (PFE) [38] and Hedge Image Embedding (HIB) [33] perform amortized inference and thus estimate the mean and variance of latent observation. We also compare against MC Dropout [13] and Deep Ensemble [23], two approximate Bayesian methods, which have successfully been applied in image retrieval [43, 42].

We compare the models' *predictive performance* with the recall (recall@$k$) and mean average precision (mAP@$k$) among the $k$ nearest neighbors [47, 30, 1]. We evaluate the models' abilities to *interpolate* and *extrapolate* uncertainties by measuring the Area Under the Sparsification Curve (AUSC), Expected Calibration Error (ECE) on in-distribution (ID) data, the Area Under Receiver Operator Curve (AUROC), and Area Under Precision-Recall Curve (AUPRC) on out-of-distribution (OoD) data. We provide more details on these metrics in **??**.

We extend StochMan [11] with the Hessian backpropagation for the contrastive loss, and the PyTorch [34] code is publicly available[2]. **??** details the experimental setup.

---

[2]See `https://github.com/FrederikWarburg/bayesian-metric-learning`

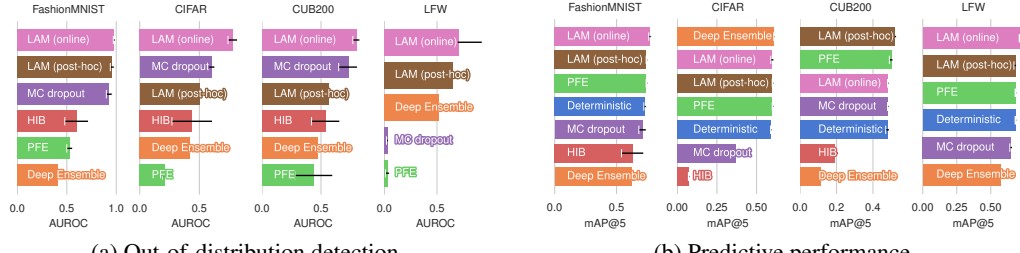

(a) Out-of-distribution detection.

(b) Predictive performance.

Figure 4: **Summary of experimental results.** LAM consistently outperforms existing methods on OoD detection, as measured by AUROC, and matches or surpasses in predictive performance measured by mAP@$k$. Error bars show one std across five runs.

**Experimental Summary.** We first summarize our experimental results. Across five datasets, three network architectures, and three different sizes of the latent space (ranging from 3 to 2048), we find that LAM has well-calibrated uncertainties, reliably detects OoD examples, and achieves state-of-the-art predictive performance. Fig. 4a shows that the uncertainties from online LAM reliably identify OoD examples. Online LAM outperforms other Bayesian methods, such as post-hoc LAM and MC dropout, on this task, which in turn clearly improves upon amortized methods that rely on a neural network to extrapolate uncertainties. Fig. 4b shows that LAM consistently matches or outperforms existing image retrieval methods in terms of predictive performance. We find that the fixed Hessian approximation with the Arccos distance performs the best, especially on higher dimensional data.

**Ablation: Positive definiteness covariance matrix.** We experimentally study which method to ensure a positive semidefinite Hessian has the best performance measured in both predictive performance (mAP@5) and uncertainty quantification (AUROC, AUSC). We found that all methods perform similarly on simple datasets and low dimensional hyper-spheres, but the fixed approximation with Arccos distance performs better on more challenging datasets and higher dimensional hyper-spheres. We present results on one of these more challenging datasets, namely the LFW [17] face recognition dataset with the CUB200 [45] bird dataset as an OoD dataset. We use a ResNet50 [16] with a GeM pooling layer [35] and a 2048 dimensional embedding and diagonal, last-layer LA [8].

Table 1 shows the performance for post-hoc and online LA with fixed, positive, or full Hessian approximation using either Euclidean or Arccos distance. Across all metrics, the online LA with Arccos distance and the fixed Hessian approximation performs similarly or the best. We proceed to benchmark this method against several strong probabilistic baselines on closed-set retrieval and a more challenging open-set retrieval.

Table 1: **Ablation on Hessian approximation and GGN decomposition.** Online LA with the fixed approximation and Arccos distance performs best. Error bars show one std. across five runs.

|          |               | mAP@5 ↑ | AUROC ↑ | AUSC ↑ |
|----------|---------------|---------|---------|--------|
| Post-hoc | Euclidean fix | $0.70 \pm 0.0$ | $0.57 \pm 0.25$ | $0.44 \pm 0.01$ |
|          | Euclidean pos | $0.70 \pm 0.0$ | $0.58 \pm 0.23$ | $0.45 \pm 0.01$ |
|          | Euclidean full | $0.70 \pm 0.0$ | $0.56 \pm 0.26$ | $0.44 \pm 0.01$ |
|          | Arccos fix    | $0.69 \pm 0.0$ | $0.53 \pm 0.20$ | $0.46 \pm 0.02$ |
|          | Arccos pos    | $0.70 \pm 0.0$ | $0.29 \pm 0.11$ | $0.48 \pm 0.01$ |
|          | Arccos full   | $0.69 \pm 0.0$ | $0.55 \pm 0.18$ | $0.45 \pm 0.01$ |
| Online   | Euclidean fix | $0.63 \pm 0.01$ | $0.77 \pm 0.04$ | $0.31 \pm 0.02$ |
|          | Euclidean pos | $0.70 \pm 0.0$ | $0.38 \pm 0.10$ | $0.47 \pm 0.01$ |
|          | Euclidean full | $0.67 \pm 0.01$ | $0.59 \pm 0.04$ | $0.42 \pm 0.01$ |
|          | Arccos fix    | $\mathbf{0.71 \pm 0.0}$ | $\mathbf{0.78 \pm 0.18}$ | $0.50 \pm 0.03$ |
|          | Arccos pos    | $0.70 \pm 0.0$ | $0.23 \pm 0.03$ | $0.46 \pm 0.00$ |
|          | Arccos full   | $\mathbf{0.71 \pm 0.0}$ | $0.70 \pm 0.12$ | $\mathbf{0.51 \pm 0.01}$ |

(a) FashionMNIST

(b) CIFAR10

Figure 5: **Calibration Curves.**

**Closed-Set Retrieval.** OoD capabilities are critical for identifying distributional shifts, outliers, and irregular user inputs, which might hinder the propagation of erroneous decisions in an automated system. We evaluate OoD performance on the commonly used benchmarks [31], where we use (1)

Table 2: **Closed-set results.** LAM matches or outperforms existing methods in terms of predictive performance. It produces reliable uncertainties ID and OoD on two standard datasets FashionMNIST and CIFAR10. Error bars show one std. across five runs.

| | | IMAGE RETRIEVAL | | | OoD | | CALIBRATION | |
|---|---|---|---|---|---|---|---|---|
| | | mAP@1 ↑ | mAP@5 ↑ | mAP@10 ↑ | AUROC ↑ | AUPRC ↑ | AUSC ↑ | ECE ↓ |
| **FashionMNIST** | Deterministic | $0.78 \pm 0.01$ | $0.73 \pm 0.01$ | $0.72 \pm 0.01$ | — | — | — | — |
| | Deep Ensemble | 0.69 | 0.62 | 0.59 | 0.41 | 0.46 | 0.61 | 0.04 |
| | PFE | $0.78 \pm 0.00$ | $0.74 \pm 0.00$ | $0.72 \pm 0.00$ | $0.53 \pm 0.03$ | $0.46 \pm 0.01$ | $0.65 \pm 0.01$ | $0.26 \pm 0.02$ |
| | HIB | $0.69 \pm 0.08$ | $0.63 \pm 0.09$ | $0.61 \pm 0.09$ | $0.60 \pm 0.12$ | $0.60 \pm 0.11$ | $0.65 \pm 0.08$ | $0.54 \pm 0.08$ |
| | MC dropout | $0.76 \pm 0.03$ | $0.71 \pm 0.03$ | $0.70 \pm 0.03$ | $0.93 \pm 0.03$ | $0.93 \pm 0.03$ | $0.84 \pm 0.06$ | $0.03 \pm 0.04$ |
| | LAM (post-hoc) | $0.78 \pm 0.00$ | $0.74 \pm 0.00$ | $0.72 \pm 0.00$ | $0.96 \pm 0.02$ | $0.96 \pm 0.02$ | $0.86 \pm 0.01$ | $0.03 \pm 0.00$ |
| | LAM (online) | $\mathbf{0.81 \pm 0.00}$ | $\mathbf{0.77 \pm 0.01}$ | $\mathbf{0.76 \pm 0.01}$ | $\mathbf{0.98 \pm 0.01}$ | $\mathbf{0.98 \pm 0.01}$ | $\mathbf{0.89 \pm 0.01}$ | $\mathbf{0.02 \pm 0.00}$ |
| **CIFAR10** | Deterministic | $\mathbf{0.66 \pm 0.00}$ | $0.59 \pm 0.00$ | $0.58 \pm 0.00$ | — | — | — | — |
| | Deep Ensemble | $\mathbf{0.66}$ | $\mathbf{0.61}$ | $\mathbf{0.59}$ | 0.42 | 0.67 | 0.72 | 0.02 |
| | MC dropout | $0.46 \pm 0.01$ | $0.37 \pm 0.01$ | $0.34 \pm 0.01$ | $0.60 \pm 0.03$ | $0.76 \pm 0.02$ | $0.61 \pm 0.01$ | $0.05 \pm 0.00$ |
| | HIB | $0.11 \pm 0.01$ | $0.07 \pm 0.00$ | $0.05 \pm 0.00$ | $0.44 \pm 0.17$ | $0.70 \pm 0.1$ | $0.29 \pm 0.03$ | $0.04 \pm 0.02$ |
| | PFE | $\mathbf{0.66 \pm 0.00}$ | $0.60 \pm 0.00$ | $0.58 \pm 0.00$ | $0.21 \pm 0.02$ | $0.56 \pm 0.01$ | $0.56 \pm 0.01$ | $0.11 \pm 0.01$ |
| | LAM (post-hoc) | $\mathbf{0.66 \pm 0.00}$ | $0.60 \pm 0.00$ | $0.58 \pm 0.00$ | $0.50 \pm 0.11$ | $0.69 \pm 0.07$ | $0.81 \pm 0.01$ | $0.23 \pm 0.01$ |
| | LAM (online) | $\mathbf{0.66 \pm 0.01}$ | $0.60 \pm 0.00$ | $0.57 \pm 0.01$ | $\mathbf{0.78 \pm 0.04}$ | $\mathbf{0.85 \pm 0.03}$ | $\mathbf{0.83 \pm 0.01}$ | $\mathbf{0.01 \pm 0.00}$ |

FashionMNIST [49] as ID and MNIST [26] as OoD, and (2) CIFAR10 [21] as ID and SVHN [32] as OoD. We use, respectively, a standard 2- or 3-layer relu convolutional network followed by a single linear layer on which we compute LA with a diagonal Hessian.

*FashionMNIST (ID) vs MNIST (OoD).* Table 2 shows that both PFE and post-hoc LAM have a similar predictive performance to the deterministic model. This is not surprising, as both methods are initialized with the deterministic parameters, and then uncertainties are learned (PFE) or deduced (post-hoc LAM) with frozen weights. The awareness of uncertainties during training, grants the online LAM slightly higher predictive performance.

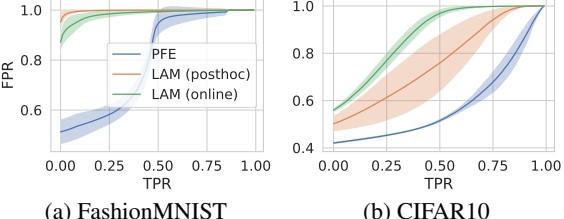

(a) FashionMNIST    (b) CIFAR10

Figure 6: **Receiver Operator Curves.** LAM assign high uncertainty to OoD observations.

PFE uses amortized inference to predict variances. This works reasonably within distribution but does not work well for OoD detection. This is because a neural network is trusted to extrapolate far away from the data distribution. Table 2 shows that MC dropout, LAM (post-hoc), and LAM (online) assign high uncertainty to observations outside the training distribution. Fig. 5 shows that both post-hoc and online LAM are near perfectly calibrated, giving very low ECE measures (Table 2).

*CIFAR10 (ID) vs SVHN (OoD)* is a slightly harder setting. Table 2 yields similar conclusions as before; Bayesian approaches such as MC dropout, LAM (post-hoc), and LAM (online) better detect OoD examples than neural amortized methods such as PFE. Online LAM has a similar predictive performance to state-of-the-art while having better ID (lower ECE and higher AUSC) and OoD (higher AUROC and AUPRC) performance. Fig. 5 shows the calibration plot for CIFAR10, where online LAM has near-perfect calibration. The CIFAR10 ROC curves (Fig. 6) show that online LAM is better at distinguishing ID and OoD examples.

**Open-Set Retrieval.** A key advantage of metric learning methods is that they easily cope with a large number of classes and new classes can be added seamlessly. We therefore evaluate LAM's performance on challenging open-set retrieval, where none of the classes in the test set are available during training. We first test with CUB200 [45] as ID and CAR196 [20] as OoD similarly to Warburg et al. [47], and second, test with LFW [17] as ID and CUB200 as OoD. We use a ResNet50 [16] with a GeM pooling layer [35] and a 2048 dimensional embedding and diagonal, last-layer LA [8].

*CUB200 (ID) vs CARS196 (OoD).* The CUB-200-2011 dataset [45] has 200 bird species captured from different perspectives and in different environments. We follow the zero-shot train/test split [30]. In this zero-shot setting, the trained models have not seen any of the bird species in the test set, and the learned features must generalize well across species. Table 3 shows that LAM matches or surpasses the predictive performance of all other methods. LAM (post-hoc) achieves state-of-the-art

Table 3: **Open-set results.** LAM matches or outperforms existing methods in terms of predictive performance and produces state-of-the-art uncertainty quantification for challenging zero-shot metric learning datasets LFW and CUB200. Error bars show one std. across five runs.

| | | IMAGE RETRIEVAL | | | OoD | | ID |
| | | mAP@1 ↑ | mAP@5 ↑ | mAP@10 ↑ | AUROC ↑ | AUPRC ↑ | AUSC ↑ |
|---|---|---|---|---|---|---|---|
| CUB200 | Deterministic | $0.62 \pm 0.01$ | $0.48 \pm 0.01$ | $0.42 \pm 0.01$ | — | — | |
| | Deep Ensemble | 0.21 | 0.11 | 0.07 | 0.47 | 0.55 | 0.21 |
| | PFE | $0.62 \pm 0.01$ | $0.5 \pm 0.01$ | $0.43 \pm 0.01$ | $0.44 \pm 0.16$ | $0.5 \pm 0.08$ | $0.61 \pm 0.02$ |
| | HIB | $0.33 \pm 0.04$ | $0.19 \pm 0.02$ | $0.14 \pm 0.02$ | $0.54 \pm 0.12$ | $0.61 \pm 0.1$ | $0.31 \pm 0.07$ |
| | MC dropout | $0.61 \pm 0.00$ | $0.48 \pm 0.00$ | $0.42 \pm 0.00$ | $0.73 \pm 0.08$ | $0.68 \pm 0.07$ | $0.63 \pm 0.01$ |
| | LAM (post-hoc) | $\mathbf{0.65 \pm 0.01}$ | $\mathbf{0.52 \pm 0.01}$ | $\mathbf{0.45 \pm 0.01}$ | $0.56 \pm 0.16$ | $0.61 \pm 0.11$ | $\mathbf{0.66 \pm 0.03}$ |
| | LAM (online) | $0.61 \pm 0.00$ | $0.48 \pm 0.00$ | $0.42 \pm 0.00$ | $\mathbf{0.80 \pm 0.03}$ | $\mathbf{0.75 \pm 0.03}$ | $0.63 \pm 0.01$ |
| LFW | Deterministic | $0.44 \pm 0.00$ | $0.68 \pm 0.00$ | $0.65 \pm 0.00$ | — | — | |
| | Deep Ensemble | 0.36 | 0.57 | 0.54 | 0.52 | 0.64 | 0.33 |
| | PFE | $0.44 \pm 0.00$ | $0.68 \pm 0.00$ | $0.65 \pm 0.00$ | $0.03 \pm 0.02$ | $0.41 \pm 0.0$ | $0.49 \pm 0.01$ |
| | MC dropout | $0.42 \pm 0.00$ | $0.65 \pm 0.01$ | $0.63 \pm 0.01$ | $0.03 \pm 0.01$ | $0.41 \pm 0.0$ | $0.46 \pm 0.01$ |
| | LAM (post-hoc) | $0.44 \pm 0.01$ | $0.68 \pm 0.01$ | $0.65 \pm 0.00$ | $0.65 \pm 0.14$ | $0.72 \pm 0.11$ | $0.45 \pm 0.03$ |
| | LAM (online) | $\mathbf{0.46 \pm 0.00}$ | $\mathbf{0.71 \pm 0.00}$ | $\mathbf{0.69 \pm 0.00}$ | $\mathbf{0.71 \pm 0.22}$ | $\mathbf{0.78 \pm 0.17}$ | $\mathbf{0.50 \pm 0.02}$ |

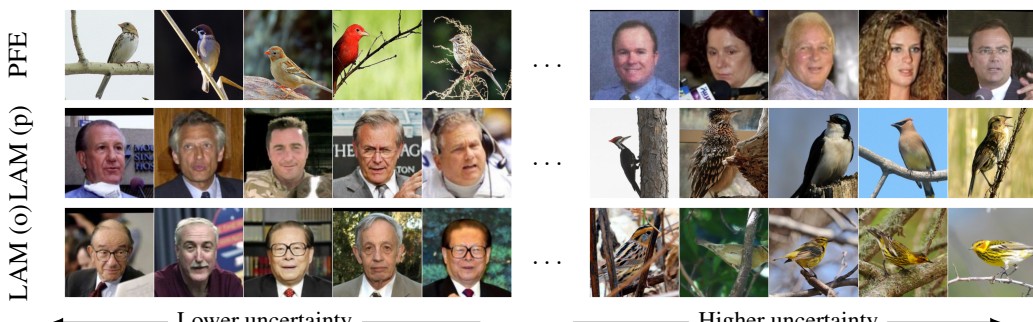

Figure 7: **Images with lowest and highest variance** for PFE, post-hoc LAM, and online LAM across LFW (ID) and CUB200 (OoD) datasets. LAM associates high uncertainty to OoD examples, and vice versa for PFE. Shows the best-performing PFE and online LAM across five runs.

predictive performance, while LAM (online) matches the predictive performance of the deterministic trained model while achieving state-of-the-art AUROC and AUPRC for OoD detection.

*LFW (ID) vs CUB200 (OoD).* Face recognition is another challenging metric learning task with many applications in security and surveillance. The goal is to retrieve images of the same person as in the query image. Table 3 shows that online LAM outperforms existing methods both in terms of predictive performance and uncertainty quantification. Fig. 7 shows that PFE assigns higher uncertainty to images from the ID dataset (faces) than those from the OoD dataset (birds). In contrast, both online and post-hoc LAM better associate high variance to OoD examples, while PFE predicts high variance to ID examples. Furthermore, online LAM seems to assign the highest variance to images in which the background is complex and thus camouflages the birds.

**Visual Place recognition** is important for the long-term operation of autonomous robots [7], where the goal is to retrieve images taken within a radius of 25 meters from a query image. The high number of unique places and varying visual appearance of each location – including weather, dynamic, structural, viewpoint, seasonal, and day/night changes – makes visual place recognition a challenging metric learning

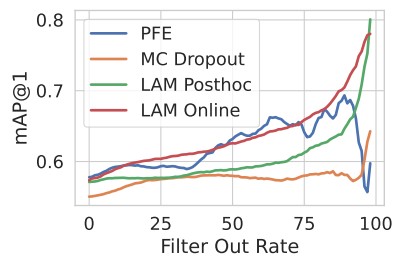

Figure 8: **Sparsification curve.** Online and post-hoc LAM's sparsification curves monotonically increase, showing that they reliably associate higher uncertainty to harder observations.

Table 5: **Results on MSLS.** LAM yields state-of-the-art uncertainties and matches the predictive performance of deterministic trained models.

| | Validation Set | | | | | | Challenge Set | | | | | |
| | R@1↑ | R@5↑ | R@10↑ | M@5↑ | M@10↑ | AUSC↑ | R@1↑ | R@5↑ | R@10↑ | M@5↑ | M@10↑ | AUSC↑ |
|---|---|---|---|---|---|---|---|---|---|---|---|---|
| Deterministic | **0.77** | **0.88** | **0.90** | **0.61** | **0.56** | — | **0.58** | **0.74** | **0.78** | **0.45** | 0.43 | — |
| MC Dropout | 0.75 | 0.87 | 0.87 | 0.59 | 0.54 | **0.77** | 0.55 | 0.71 | 0.76 | 0.43 | 0.41 | 0.57 |
| PFE | **0.77** | **0.88** | **0.90** | **0.61** | **0.56** | 0.73 | **0.58** | **0.74** | **0.78** | **0.45** | **0.44** | 0.57 |
| LAM (post-hoc) | 0.76 | 0.86 | 0.89 | 0.60 | 0.55 | 0.74 | **0.58** | **0.74** | **0.78** | **0.45** | **0.44** | 0.59 |
| LAM (online) | 0.76 | 0.87 | **0.90** | 0.60 | **0.56** | **0.77** | 0.57 | **0.74** | **0.78** | **0.45** | 0.43 | **0.63** |

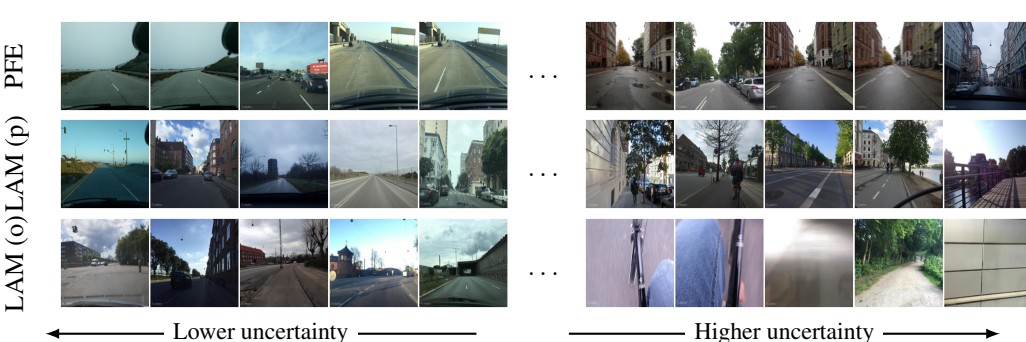

PFE · LAM (p) · LAM (o)

◄———— Lower uncertainty ————  ———— Higher uncertainty ————►

Figure 9: **Images with lowest and highest variance** for PFE, post-hoc LAM, and online LAM across MSLS validation set. LAM reliably associates high uncertainty to images that are blurry, are captured facing the pavement, or contain vegetation. These images do not contain features that are descriptive of a specific place, making them especially challenging to geographically locate.

problem. Reliable uncertainties and reliable out-of-distribution behavior are important to avoid incorrect loop-closure, which can deteriorate the autonomous robots' location estimate. We evaluate on MSLS [46], which is the largest and most diverse place recognition dataset currently available comprised of $1.6M$ images from 30 cities spanning six continents. We use the standard train/test split, training on 24 cities and testing on six other cities. We use the same model as in open-set retrieval.

Table 5 shows that online LAM yields state-of-the-art uncertainties for visual place recognition measured with AUSC, while matching the predictive performance of the alternative probabilistic and deterministic methods on both the MSLS validation and the challenge set. Fig. 8 shows the sparsification curves on the challenge set. Both online and post-hoc LAM have monotonically increasing sparsification curves, implying that when we remove the most uncertain observations, the predictive performance increase. This illustrates that LAM produces reliable uncertainties for this challenging open-set retrieval task. Fig. 9 shows the queries associated with the highest and lowest uncertainty. LAM predicts high uncertainty to images with are blurry, captured facing into the pavement, or contain mostly vegetation. These images do not have features that are descriptive of a specific place, making them hard to geographically locate.

**Ablations.** We conduct ablations on LFW for online and post-hoc LAM. Table 4 shows that the predictive performance does not depend on the margin, however, OOD performance decreases significantly for larger margins. We find that LAM is robust to the number of pairs sampled per batch. We find that a larger latent space results in better-calibrated uncertainties. Online LAM is rather robust to the choice of memory factor $\alpha$. However, choosing a memory factor in the 0.0001–0.001 range gives both good AUROC and AUSC. This memory factor $\alpha$ can be interpreted as a momentum-like parameter that relates to the learning rate and indicates how fast we should update the current Hessian [29].

| Margin | | | |
|---|---|---|---|
| | mAP@5 | AUROC | AUSC |
| 0.1 | 0.71 | 1.00 | 0.50 |
| 0.3 | 0.71 | 0.57 | 0.49 |
| 0.5 | 0.70 | 0.16 | 0.48 |
| 0.7 | 0.70 | 0.09 | 0.48 |

| Number of pairs per batch | | | |
|---|---|---|---|
| | mAP@5 | AUROC | AUSC |
| 1 | 0.72 | 0.99 | 0.49 |
| 5 | 0.72 | 0.98 | 0.50 |
| 10 | 0.71 | 1.00 | 0.49 |
| 30 | 0.72 | 1.00 | 0.49 |

| Latent dimension | | | |
|---|---|---|---|
| | mAP@5 | AUROC | AUSC |
| 128 | 0.71 | 0.99 | 0.47 |
| 256 | 0.71 | 1.00 | 0.49 |
| 512 | 0.71 | 1.00 | 0.50 |
| 2048 | 0.72 | 1.00 | 0.52 |

| Memory factor $\alpha$ | | | |
|---|---|---|---|
| | mAP@5 | AUROC | AUSC |
| 0.1 | 0.70 | 1.00 | 0.41 |
| 0.01 | 0.71 | 0.99 | 0.46 |
| 0.001 | 0.72 | 1.00 | 0.48 |
| 0.0001 | 0.72 | 0.98 | 0.50 |
| 0.00001 | 0.72 | 0.87 | 0.51 |

| Post-hoc tempering $\beta$ | | | |
|---|---|---|---|
| | mAP@5 | AUROC | AUSC |
| 100 | 0.67 | 0.10 | 0.48 |
| 10 | 0.69 | 0.68 | 0.46 |
| 1 | 0.69 | 0.86 | 0.44 |
| 0.1 | 0.69 | 0.87 | 0.43 |
| 0.01 | 0.69 | 0.86 | 0.44 |
| 0.001 | 0.69 | 0.86 | 0.43 |

Table 4: **Ablation on LFW.**

Lastly, we perform cold-posterior tempering of post-hoc LA by scaling the log-likelihood hessian with a factor $\beta > 0$, such that the precision in Eq. 3 became $\beta \cdot \nabla_\theta^2 \mathcal{L}_{\text{con}}(\theta^*; \mathcal{D}) + \sigma_{\text{prior}}^{-2} \mathbb{I}$. We do not see any benefits from changing to other values than one.

**Limitations.** Similar to other Bayesian methods, LAM relies on $n$ samples to obtain uncertainties. This makes inferences $n$ times slower. Computing the Hessian at every step during online LAM also makes training time slower. To combat long training times and high memory usage, we use a last-layer LAM and thus only estimate and sample for a weight posterior of the last layer. The last-layer LAM training time is 3 hours for online LAM vs 2.3 hours for deterministic contrastive loss on LFW, and 30 minutes vs 15 minutes loss on CUB200 on an NVIDIA RTX A5000.

## 5    Conclusion

In this paper, we have introduced a Bayesian encoder for metric learning, the Laplacian Metric Learner (LAM), which uses the Laplace approximation. We prove that the contrastive loss is unnormalized negative log-likelihood on the spherical space, and develop three Hessian approximations, which ensures a positive definite covariance matrix. We propose a novel decomposition of the Generalized Gauss-Newton approximation that improves Hessian approximations of $\ell_2$-normalized networks. Empirically, we demonstrate that LAM consistently produces well-calibrated uncertainties, reliably detects out-of-distribution examples, and achieves state-of-the-art predictive performance on both closed-set and challenging open-set image retrieval tasks.

**Acknowledgement.** This work was supported by research grants (15334, 42062) from VILLUM FONDEN. This project received funding from the European Research Council (ERC) under the European Union's Horizon 2020 research and innovation programme (grant agreement 757360). The work was partly funded by the Novo Nordisk Foundation through the Center for Basic Machine Learning Research in Life Science (NNF20OC0062606).

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
