# Supplementary Material: Bayesian Metric Learning for Uncertainty Quantification in Image Retrieval

**Frederik Warburg**[*]
Technical University of Denmark
frwa@dtu.dk

**Marco Miani**[*]
Technical University of Denmark
mmia@dtu.dk

**Silas Brack**
Technical University of Denmark
silasbrack@gmail.com

**Søren Hauberg**
Technical University of Denmark
sohau@dtu.dk

## A   Notation

Let $\mathcal{X}$ be the data space, $\mathcal{Z}$ be the latent space, $\Theta$ be the parameter space. For a given neural network architecture $f$ and a parameter vector $\theta \in \Theta$, $f_\theta$ is a function

$$f_\theta : \mathcal{X} \to \mathcal{Z} \tag{1}$$

The discrete set $\mathcal{C} = \{c_i\}_{i=1,\ldots,C}$ of classes induces a partition of the points in the data space. A dataset is a collection of *data point*-class pairs $(x, c) \in \mathcal{X} \times \mathcal{C}$.

$$\mathcal{D} = \{(x_i, c_i)\}_{i=1,\ldots,D} \tag{2}$$

and it is itself partitioned by the classes into $C$ sets

$$\mathcal{D}_c = \{(x', c') \in \mathcal{D} | c' = c\} \subset \mathcal{D} \qquad \forall c \in \mathcal{C}. \tag{3}$$

We assume these sets always to contain at least one element $\mathcal{D}_c \neq \emptyset$ and we use the notation $|\mathcal{D}_c|$ to refer to their cardinalities. In the metric learning setting, instead of enforcing properties of a single data point, the goal is to enforce relations between data points. Thus we will often make use of pairs $\mathfrak{p}_{ij} = ((x_i, c_i), (x_j, c_j)) \in \mathcal{D}^2$, specifically we use the terms

$$\mathfrak{p}_{ij} \text{ is } \textit{positive} \text{ pair if } c_i = c_j \tag{4}$$

$$\mathfrak{p}_{ij} \text{ is } \textit{negative} \text{ pair if } c_i \neq c_j. \tag{5}$$

To ease the later notation we will consider the trivial pairs composed by a data point with itself. We then define the positive set and negative set, and compute their cardinalities, respectively

$$\mathcal{D}^2_{\text{pos}} := \{\mathfrak{p} \in \mathcal{D}^2 \text{ such that } p \text{ is positive}\} \qquad |\mathcal{D}^2_{\text{pos}}| = \sum_{c \in \mathcal{C}} |\mathcal{D}_c|^2 \tag{6}$$

$$\mathcal{D}^2_{\text{neg}} := \{\mathfrak{p} \in \mathcal{D}^2 \text{ such that } p \text{ is negative}\} \qquad |\mathcal{D}^2_{\text{neg}}| = \sum_{\substack{c_i, c_j \in \mathcal{C}^2 \\ \text{s.t. } c_i \neq c_j}} |\mathcal{D}_{c_i}||\mathcal{D}_{c_j}| \tag{7}$$

A common trick in metric learning is to introduce a *margin* $m \in \mathbb{R}^+$. The margin induces a further partition of the $\mathcal{D}^2_{\text{neg}}$ set into pairs with close or far embeddings. For any pair $\mathfrak{p}_{ij} = ((x_i, c_i), (x_j, c_j))$ we describe the pair as being

$$\textit{inside} \text{ the margin if } \|f_\theta(x_i) - f_\theta(x_j)\| \leq m \tag{8}$$

$$\textit{outside} \text{ the margin if } \|f_\theta(x_i) - f_\theta(x_j)\| > m \tag{9}$$

---

[*]Equal contribution

37th Conference on Neural Information Processing Systems (NeurIPS 2023).

irrespective of the classes $c_i$ and $c_j$. We can make use of this definition to consider only the pairs that have a non-zero loss. From this logic, it is convenient to define the set $\mathcal{D}^2_{\text{neg inside}} := \{ \mathfrak{p} \in \mathcal{D}^2_{\text{neg}} \text{ inside the margin} \}$.

The *target*, or label, is the value that encodes the information we want to learn. In classical settings, we have one scalar for each data point: a class for classification, a value for regression. In our setting, we consider a target $y_{ij} \in \mathbb{R}$ for every pair $\mathfrak{p}_{ij} \in \mathcal{D}^2$. With this notation in mind, we can continue to define the contrastive loss.

## B  Contrastive Loss

Recall the definition of the contrastive loss [1]

$$\mathcal{L}_{\text{con}}(\theta) = \frac{1}{2}\|f_\theta(x_a) - f_\theta(x_p)\|^2 + \frac{1}{2}\max\left(0, m - \|f_\theta(x_a) - f_\theta(x_n)\|^2\right), \tag{10}$$

where $f$ is a network with parameters $\theta$ and $x_p$ is a data point the same class as the anchor $x_a$ and different from the negative data point $x_n$. The loss over the whole dataset is then informally defined as $\mathcal{L}_{\text{con}}(\theta; \mathcal{D}) = \mathbb{E}_\mathcal{D}[\mathcal{L}_{\text{con}}(\theta)]$, where the expectation is taken over tuples $(x_a, x_p, x_n)$ satisfying the positive and negative constrains. This definition is intuitive and compact, but not formal enough to show that the contrastive loss is in fact an unnormalized log posterior. The main issue is the explicit notation for positive $x_p$ and negative $x_n$, which at first glance seems innocent, but for later derivations becomes cumbersome. Instead, we express the loss in an equivalent but more verbose way.

In order to understand that we are doing nothing more than a change in notation, it is first convenient to express Eq. 10 as

$$\mathcal{L}_{\text{con}}(\theta) = \begin{cases} \frac{1}{2}\|f_\theta(x_i) - f_\theta(x_j)\|^2 & \text{if the pair } \mathfrak{p}_{ij} \text{ is positive} \\ 0 & \text{if the pair } \mathfrak{p}_{ij} \text{ is negative and outside the margin} \\ m - \frac{1}{2}\|f_\theta(x_i) - f_\theta(x_j)\|^2 & \text{if the pair } \mathfrak{p}_{ij} \text{ is negative and inside the margin} \end{cases}, \tag{11}$$

which reveals the three cases of the *per-observation* contrastive loss. Notice that, up to a neglectable additive constant, all cases shares the same form of being scalar multiples of a distance between the two embeddings $z_i = f_\theta(x_i)$ and $z_j = f_\theta(x_j)$. This allows us to combine the three scenarios into a single-case *per-observation* contrastive loss, parametrized by a scalar $y \in \mathbb{R}$, as

$$\begin{aligned} \mathcal{L}_y: \quad \mathcal{Z}^2 &\longrightarrow \mathbb{R} \\ z_i, z_j &\longmapsto \frac{1}{2}y\|z_i - z_j\|^2 \end{aligned} \tag{12}$$

and we later entrust the distinction between the three scenarios, as in Eq. 11, to the scalar $y$. Although it is important to define this loss for a general target $y \in \mathbb{R}$ (we will interpret it as a Von-Mises-Fisher concentration parameter $\kappa$ in Appendix D), practically we will make use of the specific instances $y = y_{ij} \in \mathbb{R}$ with the target values defined for every data indexes $i$ and $j$ as

$$y_{ij} := \begin{cases} \frac{1}{|\mathcal{D}^2_{\text{pos}}|} & \text{if the pair } \mathfrak{p}_{ij} \text{ is positive} \\ 0 & \text{if the pair } \mathfrak{p}_{ij} \text{ is negative and outside the margin} \\ -\frac{1}{|\mathcal{D}^2_{\text{neg}}|} & \text{if the pair } \mathfrak{p}_{ij} \text{ is negative and inside the margin} \end{cases}. \tag{13}$$

We define the contrastive loss over the entire dataset $\mathcal{L}(\ ; \mathcal{D}) : \Theta \to \mathbb{R}$ as a sum over all pairs of the *per-observation* contrastive loss (12) with targets (13)

$$\mathcal{L}(\theta; \mathcal{D}) := \sum_{\mathfrak{p}_{ij} \in \mathcal{D}^2} \mathcal{L}_{y_{ij}}(\underbrace{f_\theta(x_i)}_{z_i}, \underbrace{f_\theta(x_j)}_{z_j}). \tag{14}$$

Notice that we slightly overload the notation $\mathcal{L}$ and $\mathcal{L}_y$ for the dataset loss and the per-observation-loss, respectively.

**Expanded expression**. In order to better highlight the equivalence, it may be useful to express the loss explicitly using all the previous definitions.

$$\mathcal{L}(\theta; \mathcal{D}) = \sum_{\mathfrak{p}_{ij} \in \mathcal{D}^2} \mathcal{L}_{y_{ij}}(f_\theta(x_i), f_\theta(x_j)) \tag{15}$$

$$= \sum_{\mathfrak{p}_{ij} \in \mathcal{D}^2} \frac{1}{2} y_{ij} \|f_\theta(x_i) - f_\theta(x_j)\|^2 \tag{16}$$

$$= \sum_{\mathfrak{p}_{ij} \in \mathcal{D}^2_{\text{pos}}} \frac{1}{2} \frac{1}{|\mathcal{D}^2_{\text{pos}}|} \|f_\theta(x_i) - f_\theta(x_j)\|^2 + \sum_{\mathfrak{p}_{ij} \in \mathcal{D}^2_{\text{neg inside}}} -\frac{1}{2} \frac{1}{|\mathcal{D}^2_{\text{neg}}|} \|f_\theta(x_i) - f_\theta(x_j)\|^2 \tag{17}$$

$$= \frac{1}{|\mathcal{D}^2_{\text{pos}}|} \sum_{\mathfrak{p}_{ij} \in \mathcal{D}^2_{\text{pos}}} \frac{1}{2} \|f_\theta(x_i) - f_\theta(x_j)\|^2 - \frac{1}{|\mathcal{D}^2_{\text{neg}}|} \sum_{\mathfrak{p}_{ij} \in \mathcal{D}^2_{\text{neg inside}}} \frac{1}{2} \|f_\theta(x_i) - f_\theta(x_j)\|^2 \tag{18}$$

The scaling factor $\frac{1}{|\mathcal{D}^2_{\text{pos}}|}$ for positives and $\frac{1}{|\mathcal{D}^2_{\text{neg}}|}$ for negatives, together with the sum over pairs, leads to a per-type average. In this sense, we can informally say that $\mathcal{L}(\theta) = \mathbb{E}[\mathcal{L}_{\text{con}}(\theta)]$ under a distribution that assigns equal probabilities of a pair being positive or negative. We highlight that scaling is key in order to ensure positive definiteness in Proposition F.3.

## B.1   Minibatch

In the previous Section, we defined the contrastive loss for the entire dataset (14). However, in practice it is approximated with minibatching. This approximation induces a bias that we can correct by simply adjusting the target $y_{ij}$ as in Eq. 20.

**Minibatching recap.** When dealing with a huge number of identically distributed data, it is common to assume that a big enough arbitrary subset will follow the same distribution and thus have the same properties, specifically we assume the expected value to be similar $\mathbb{E}_{\mathcal{S}}[f(s)] \approx \mathbb{E}_{\mathcal{S}'}[f(s)]$. The idea of minibatching relies on this to approximate a sum over a set $\mathcal{S}$ with a *scaled* sum over a subset $\mathcal{S}' \subseteq \mathcal{S}$,

$$\sum_{s \in \mathcal{S}} f(s) \approx \frac{|\mathcal{S}|}{|\mathcal{S}'|} \sum_{s \in \mathcal{S}'} f(s) \tag{19}$$

where $\mathcal{S} = \mathcal{D}$ and $\mathcal{S}'$ is the set of data points in the minibatch.

**Minibatching contrastive.** Conversely, in the constrastive setting, we need to consider a subset of the pairs (rather than single observations), i.e. $\mathcal{S} = \mathcal{D}^2$. In practice this subset will not be representative of the positive and negative ratio, thus we need a different scaling to account for that. This process can be viewed as taking two independent minibatches at the same time, one representative of the positives and one of the negatives. This intuition is formalized by the following Definition and Proposition.

**Definition.** Consider a minibatch set of pairs $\mathcal{B} \subseteq \mathcal{D}^2$ and its partition $\mathcal{B} = \mathcal{B}_{\text{pos}} \cup \mathcal{B}_{\text{neg}}$ in positives $\mathcal{B}_{\text{pos}} \subseteq \mathcal{D}^2_{\text{pos}}$ and negatives $\mathcal{B}_{\text{neg}} \subseteq \mathcal{D}^2_{\text{neg}}$, then we can define a new scaled target as

$$y^{\mathcal{B}}_{ij} := \begin{cases} \frac{|\mathcal{B}|}{|\mathcal{D}^2||\mathcal{B}_{\text{pos}}|} & \text{if the pair } \mathfrak{p}_{ij} \text{ is positive} \\ 0 & \text{if the pair } \mathfrak{p}_{ij} \text{ is negative and outside the margin} \\ -\frac{|\mathcal{B}|}{|\mathcal{D}^2||\mathcal{B}_{\text{neg}}|} & \text{if the pair } \mathfrak{p}_{ij} \text{ is negative and inside the margin} \end{cases} \tag{20}$$

and so we can properly approximate the loss by minibatching positives and negatives independently

**Proposition B.1.** *Assume that the positives in the batch are representative of the positives in the whole dataset, and similarly for the negatives, i.e. $\mathbb{E}_{\mathcal{D}^2_{\text{pos}}}[\mathcal{L}_y] \approx \mathbb{E}_{\mathcal{B}_{\text{pos}}}[\mathcal{L}_y]$ and $\mathbb{E}_{\mathcal{D}^2_{\text{neg}}}[\mathcal{L}_y] \approx \mathbb{E}_{\mathcal{B}_{\text{neg}}}[\mathcal{L}_y]$. Then the loss, as defined in Eq. 14, can be approximated by using the target in Eq. 20 with*

$$\mathcal{L}(\theta; \mathcal{D}) \approx \frac{|\mathcal{D}^2|}{|\mathcal{B}|} \sum_{\mathfrak{p}_{ij} \in \mathcal{B}} \mathcal{L}_{y^{\mathcal{B}}_{ij}}(f_\theta(x_i), f_\theta(x_j)) \tag{21}$$

*Proof.* The equality is proven by applying the logic of Eq. 19 two times independently, once for the positive pairs with $\mathcal{B}_{\text{pos}} = \mathcal{S}' \subseteq \mathcal{S} = \mathcal{D}^2_{\text{pos}}$ and once for the negatives with $\mathcal{B}_{\text{neg}} = \mathcal{S}' \subseteq \mathcal{S} = \mathcal{D}^2_{\text{neg}}$

and then rearranging the terms

$$\mathcal{L}(\theta; \mathcal{D}) = \sum_{\mathfrak{p}_{ij} \in \mathcal{D}^2} \mathcal{L}_{y_{ij}}(f_\theta(x_i), f_\theta(x_j)) \tag{22}$$

$$= \sum_{\mathfrak{p}_{ij} \in \mathcal{D}^2_{\text{pos}}} \mathcal{L}_{y_{ij}}(f_\theta(x_i), f_\theta(x_j)) + \sum_{\mathfrak{p}_{ij} \in \mathcal{D}^2_{\text{neg}}} \mathcal{L}_{y_{ij}}(f_\theta(x_i), f_\theta(x_j)) \tag{23}$$

$$\approx \frac{|\mathcal{D}^2_{\text{pos}}|}{|\mathcal{B}_{\text{pos}}|} \sum_{\mathfrak{p}_{ij} \in \mathcal{B}_{\text{pos}}} \mathcal{L}_{y_{ij}}(f_\theta(x_i), f_\theta(x_j)) + \frac{|\mathcal{D}^2_{\text{neg}}|}{|\mathcal{B}_{\text{neg}}|} \sum_{\mathfrak{p}_{ij} \in \mathcal{B}_{\text{neg}}} \mathcal{L}_{y_{ij}}(f_\theta(x_i), f_\theta(x_j)) \tag{24}$$

$$= \frac{|\mathcal{D}^2|}{|\mathcal{B}|} \left( \frac{|\mathcal{D}^2_{\text{pos}}|}{|\mathcal{D}^2|} \frac{|\mathcal{B}|}{|\mathcal{B}_{\text{pos}}|} \sum_{\mathfrak{p}_{ij} \in \mathcal{B}_{\text{pos}}} \mathcal{L}_{y_{ij}}(f_\theta(x_i), f_\theta(x_j)) + \frac{|\mathcal{D}^2_{\text{neg}}|}{|\mathcal{D}^2|} \frac{|\mathcal{B}|}{|\mathcal{B}_{\text{neg}}|} \sum_{\mathfrak{p}_{ij} \in \mathcal{B}_{\text{neg}}} \mathcal{L}_{y_{ij}}(f_\theta(x_i), f_\theta(x_j)) \right) \tag{25}$$

$$= \frac{|\mathcal{D}^2|}{|\mathcal{B}|} \left( \frac{\text{pos}^\%_{\mathcal{D}^2}}{\text{pos}^\%_{\mathcal{B}}} \sum_{\mathfrak{p}_{ij} \in \mathcal{B}_{\text{pos}}} \mathcal{L}_{y_{ij}}(f_\theta(x_i), f_\theta(x_j)) + \frac{\text{neg}^\%_{\mathcal{D}^2}}{\text{neg}^\%_{\mathcal{B}}} \sum_{\mathfrak{p}_{ij} \in \mathcal{B}_{\text{neg}}} \mathcal{L}_{y_{ij}}(f_\theta(x_i), f_\theta(x_j)) \right) \tag{26}$$

$$= \frac{|\mathcal{D}^2|}{|\mathcal{B}|} \left( \sum_{\mathfrak{p}_{ij} \in \mathcal{B}_{\text{pos}}} \mathcal{L}_{y^\mathcal{B}_{ij}}(f_\theta(x_i), f_\theta(x_j)) + \sum_{\mathfrak{p}_{ij} \in \mathcal{B}_{\text{neg}}} \mathcal{L}_{y^\mathcal{B}_{ij}}(f_\theta(x_i), f_\theta(x_j)) \right) \tag{27}$$

$$= \frac{|\mathcal{D}^2|}{|\mathcal{B}|} \sum_{\mathfrak{p}_{ij} \in \mathcal{B}} \mathcal{L}_{y^\mathcal{B}_{ij}}(f_\theta(x_i), f_\theta(x_j)) \tag{28}$$

where $\text{pos}^\%$ and $\text{neg}^\%$ are the percentage of respectively positives and negatives in a given set, indicated in the subscript. $\qquad\square$

We highlight that this scaling is linear, and thus is reflected in both first and second-order derivatives. This will later become important for scaling the Hessian.

## C  Normalization layer and Von Mises–Fisher

It is common in metric learning to add a normalization layer at the end of the neural network architecture. This, besides improving performances, has an interesting geometric interpretation. Moreover, happens to be fundamental in order to interpret the loss in a probabilistic way, and consequently to apply the Laplace approximation in a meaningful way.

Adding an $l^2$-normalization layer is a practical way to enforce that all outputs lie on the unit sphere. In other words, we are assuming our latent manifold to be

$$\mathcal{Z} := S^Z \subset \mathbb{R}^{Z+1}. \tag{29}$$

We can generalize the concept of Gaussian to the unit sphere. Start from a normal distribution with isotropic covariance $\sigma^2 \mathbb{I}$ and mean $\mu$. Conditioning on $\|z\| = 1$ leads to a distribution on $S^Z$, the so-called von Mises–Fisher distribution $\mathcal{N}^S$. More succinctly, the restriction of any isotropic multivariate normal density to the unit hypersphere, gives a von Mises-Fisher density, up to normalization.

$$\mathcal{N}^S(z|\mu, \kappa) \sim \mathcal{P}(z) = c_\kappa e^{-\kappa \frac{\|z-\mu\|^2}{2}} \tag{30}$$

where, importantly, the normalization constant $c_\kappa \in \mathbb{R}$ only depends on $\kappa$ and not on $\mu$. The von Mises-Fisher distribution is parametrized with a directional mean $\mu$ and a scalar concentration parameter $\kappa$, which can be interpreted as the inverse of an isotropic covariance $\kappa = \frac{1}{\sigma^2}$.

Being both $\mu$ and $z$ of the unitary norm, the norm $\|z - \mu\|^2$ can be more efficiently computed through the scalar product $\langle z, \mu \rangle$, that is the reason it is also called Arccos distance. Moreover, we highlight that on the unit sphere, the equivalence

$$\|z - \mu\|^2 = 2 - \langle z, \mu \rangle = 4 - \|z + \mu\|^2 \tag{31}$$

holds. This will be key in proving the equivalence of the probabilistic setting from Eq. 44 to Eq. 45.

**Parameters estimators**. Having access to $N$ samples drawn from a Gaussian distribution with unknown mean and variance, it is common to compute an empirical estimate of such parameters. Various estimators exists, each satisfying different properties like being unbiased [3].

In a similar fashion, for Von-Mises-Fisher distributions we make use of two empirical estimators Sra [9] of the parameters $\mu$ and $\kappa = \frac{1}{\sigma^2}$. We compute the empirical mean direction

$$\bar{\mu} = \frac{\mu}{\bar{R}}, \qquad \bar{R} = \|\mu\|, \qquad \mu = \frac{1}{N} \sum_{i=0}^{N} z_i, \tag{32}$$

from these samples and the approximate concentration parameter

$$\bar{\kappa} = \frac{\bar{R}(D - \bar{R}^2)}{1 - \bar{R}^2}. \tag{33}$$

## D   Probabilistic view

In the metric learning framework, a dataset $\mathcal{D}$ plays two very different roles at the same time. This happens in the same spirit as, in electrostatics, a charged particle at the same time *generates* an electric field and *interact* with the existing electric field. This subtle difference, although being neglectable in the classic non-Bayesian setting, is conceptually important in the probabilistic setting. Thus, for now, we assume to be given a *generating* dataset $\mathcal{D}_G$ and an *interacting* dataset $\mathcal{D}_I$. Only after the derivations, we will set them to be equal.

For some fixed value $\kappa > 0$, each data point $x \in \mathcal{X}$ induces two Von Mises-Fisher distributions on the latent space

$$\mathcal{P}^{\rightarrow\leftarrow}(z|x,\theta) \sim \mathcal{N}^S(z|f_\theta(x), \kappa) \tag{34}$$

$$\mathcal{P}^{\leftarrow\rightarrow}(z|x,\theta) \sim \mathcal{N}^S(z| - f_\theta(x), \kappa) \tag{35}$$

one centered in the embedding point $f_\theta(x)$ and one centered in the antipodal point $-f_\theta(x)$. It is convenient to explicit the densities of these two distributions, according to Eq. 30

$$\mathcal{P}^{\rightarrow\leftarrow}(z|x,\theta) = c_\kappa \exp\left(-\kappa \frac{\|z - f_\theta(x)\|^2}{2}\right) \tag{36}$$

$$\mathcal{P}^{\leftarrow\rightarrow}(z|x,\theta) = c_\kappa \exp\left(-\kappa \frac{\|z + f_\theta(x)\|^2}{2}\right). \tag{37}$$

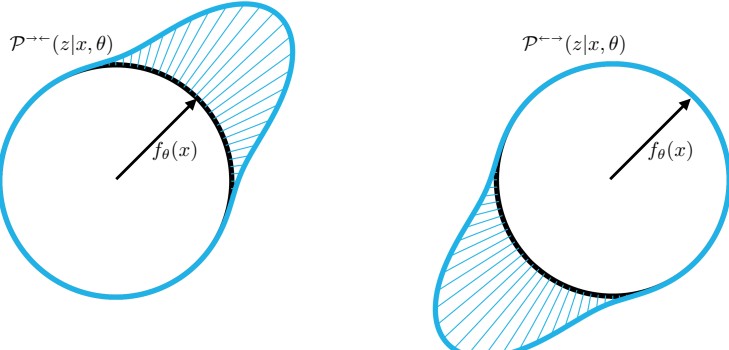

Figure 1: Representation of the Von Mises-Fisher densities $\mathcal{P}^{\rightarrow\leftarrow}$ and $\mathcal{P}^{\leftarrow\rightarrow}$ in the one-dimensional unit circle $\mathcal{Z} = \mathcal{S}^1$. Intuitively, the distribution corresponding to an attractive force is supported near the embedding $z = f_\theta(x)$, while the distribution corresponding to a repulsive force is supported far from the embedding, i.e. close to the antipodal $z = -f_\theta(x)$. The higher the value of $\kappa$ is, the higher the precision, the lower the variance and the narrower the support.

The *generating* dataset $\mathcal{D}_G$ induces a distribution $\mathcal{P}_{\mathcal{Z}}^c \in \Delta(\mathcal{Z})$ on the latent space for each class $c \in \mathcal{C}$ defined by

$$\mathcal{P}_{\mathcal{Z}}^c(z|\mathcal{D}_G, \theta) = \frac{c_\kappa}{|\mathcal{D}_G|} \prod_{\substack{(x_j, c_j) \in \mathcal{D}_G \\ \text{s.t. } c_j = c}} \mathcal{P}^{\rightarrow\leftarrow}(z|x_j, \theta) \prod_{\substack{(x_j, c_j) \in \mathcal{D}_G \\ \text{s.t. } c_j \neq c}} \mathcal{P}^{\leftarrow\rightarrow}(z|x_j, \theta) \tag{38}$$

that is a product of one *attractive* term $\mathcal{P}^{\rightarrow\leftarrow}$ for all anchor points of the same class, times a *repulsive* term $\mathcal{P}^{\leftarrow\rightarrow}$ for all anchor points of different classes. In the same spirit as the product of Gaussian densities is a Gaussian density, we can show that $\mathcal{P}_{\mathcal{Z}}^c$ is a Von Mises-Fisher distribution itself, the precision of which is the sum of the precisions. This implies that the normalization constant only depends on $\kappa$ for this distribution as well.

Having access to a likelihood in the latent space, the *interacting* dataset likelihood is then classically defined as the product

$$\mathcal{P}(\mathcal{D}_I|\mathcal{D}_G, \theta) = \prod_{(x_i, c_i) \in \mathcal{D}_I} \mathcal{P}_{\mathcal{Z}}^{c_i}(z|\mathcal{D}_G, \theta)\big|_{z = f_\theta(x_i)} \tag{39}$$

Using the definition in Eq. 38 and rearranging the terms, we can rewrite this likelihood in extended form as

$$\mathcal{P}(\mathcal{D}_I|\mathcal{D}_G, \theta) = \bar{c}_\kappa \prod_{\substack{\mathfrak{p}_{ij} \in \mathcal{D}_I \times \mathcal{D}_G \\ \text{s.t. } \mathfrak{p}_{ij} \text{ is positive}}} \mathcal{P}^{\rightarrow\leftarrow}(z|x_j, \theta)\big|_{z = f_\theta(x_i)} \prod_{\substack{\mathfrak{p}_{ij} \in \mathcal{D}_I \times \mathcal{D}_G \\ \text{s.t. } \mathfrak{p}_{ij} \text{ is negative}}} \mathcal{P}^{\leftarrow\rightarrow}(z|x_j, \theta)\big|_{z = f_\theta(x_i)} \tag{40}$$

Setting the generating and interacting dataset to be the same, we can define the constrastive learning dataset likelihood as

$$\mathcal{P}(\mathcal{D}|\theta) := \mathcal{P}(\mathcal{D}_I|\mathcal{D}_G, \theta)\big|_{\mathcal{D}_I = \mathcal{D}_G = \mathcal{D}} \tag{41}$$

### D.1 Equivalence of the two settings

The contrastive term for a single pair is equivalent to the negative log-likelihood of a von Mises-Fisher distribution, up to additive constants. Specifically, a positive target $y = \kappa > 0$ is related to the attractive term $\mathcal{P}^{\rightarrow\leftarrow}$

$$\mathcal{L}_\kappa(f_\theta(x_e), f_\theta(x_a)) = \frac{1}{2}\kappa\|f_\theta(x_e) - f_\theta(x_a)\|^2 \tag{42}$$

$$= \log(c_\kappa) - \log\mathcal{P}^{\rightarrow\leftarrow}(f_\theta(x_e)|x_a, \theta) \tag{43}$$

while a negative target $y = -\kappa < 0$ is related to the repulsive term $\mathcal{P}^{\leftarrow\rightarrow}$

$$\mathcal{L}_{-\kappa}(f_\theta(x_e), f_\theta(x_a)) = -\frac{1}{2}\kappa\|f_\theta(x_e) - f_\theta(x_a)\|^2 \tag{44}$$

$$= -2\kappa + \frac{1}{2}\kappa\|f_\theta(x_e) + f_\theta(x_a)\|^2 \tag{45}$$

$$= -2\kappa + \log(c_\kappa) - \log\mathcal{P}^{\leftarrow\rightarrow}(f_\theta(x_e)|x_a, \theta). \tag{46}$$

This equivalence is reflected in the equivalence between the contrastive loss and the dataset negative log-likelihood, up to an additive constant. Expanding the definition of dataset likelihood we have

$$\mathcal{P}(\mathcal{D}|\theta) = \prod_{(x_i, c_i) \in \mathcal{D}} \mathcal{P}_{\mathcal{Z}}^{c_i}(f_\theta(x_i)|\mathcal{D}, \theta) \tag{47}$$

$$= \bar{c}_\kappa \prod_{\substack{\mathfrak{p}_{ij} \in \mathcal{D}^2 \\ \text{s.t. } \mathfrak{p}_{ij} \text{ is positive}}} \mathcal{P}^{\rightarrow\leftarrow}(z|x_j, \theta)\big|_{z = f_\theta(x_i)} \prod_{\substack{\mathfrak{p}_{ij} \in \mathcal{D}^2 \\ \text{s.t. } \mathfrak{p}_{ij} \text{ is negative}}} \mathcal{P}^{\leftarrow\rightarrow}(z|x_j, \theta)\big|_{z = f_\theta(x_i)} \tag{48}$$

$$= \bar{c}_\kappa c_\kappa^{|\mathcal{D}|} \prod_{\substack{\mathfrak{p}_{ij} \in \mathcal{D}^2 \\ \text{s.t. } \mathfrak{p}_{ij} \text{ is positive}}} \exp\left(-\kappa\frac{\|f_\theta(x_i) - f_\theta(x_j)\|^2}{2}\right) \prod_{\substack{\mathfrak{p}_{ij} \in \mathcal{D}^2 \\ \text{s.t. } \mathfrak{p}_{ij} \text{ is negative}}} \exp\left(-\kappa\frac{\|f_\theta(x_i) + f_\theta(x_j)\|^2}{2}\right)$$
$$\tag{49}$$

Considering the log-likelihood

$$\log \mathcal{P}(\mathcal{D}|\theta) = \sum_{(x_i, c_i) \in \mathcal{D}} \log \mathcal{P}_{\mathcal{Z}}^{c_i}(f_\theta(x_i)|\mathcal{D}, \theta) \tag{50}$$

$$= \log(\bar{c}_\kappa) + \sum_{\substack{\mathfrak{p}_{ij} \in \mathcal{D}^2 \\ \text{s.t. } \mathfrak{p}_{ij} \text{ is positive}}} \log \mathcal{P}^{\rightarrow\leftarrow}(z|x_j, \theta)\big|_{z=f_\theta(x_i)} + \sum_{\substack{\mathfrak{p}_{ij} \in \mathcal{D}^2 \\ \text{s.t. } \mathfrak{p}_{ij} \text{ is negative}}} \log \mathcal{P}^{\leftarrow\rightarrow}(z|x_j, \theta)\big|_{z=f_\theta(x_i)} \tag{51}$$

$$= \text{const}(\kappa) - \sum_{\substack{\mathfrak{p}_{ij} \in \mathcal{D}^2 \\ \text{s.t. } \mathfrak{p}_{ij} \text{ is positive}}} \mathcal{L}_\kappa(f_\theta(x_i), f_\theta(x_j)) - \sum_{\substack{\mathfrak{p}_{ij} \in \mathcal{D}^2 \\ \text{s.t. } \mathfrak{p}_{ij} \text{ is negative}}} \mathcal{L}_{-\kappa}(f_\theta(x_i), f_\theta(x_j)) \tag{52}$$

$$= \text{const}(\kappa) - \mathcal{L}(\theta; \mathcal{D}) \tag{53}$$

thus the loss is equal, up to an additive constant, to the negative log-likelihood and we can proceed with the Bayesian interpretation. We highlight that $\text{const}(\kappa)$ is not dependent on $\theta$, and so is neglected by the derivatives

$$\nabla_\theta \log \mathcal{P}(\mathcal{D}|\theta) = -\nabla_\theta \mathcal{L}(\theta; \mathcal{D}) \tag{54}$$

$$\nabla_\theta^2 \log \mathcal{P}(\mathcal{D}|\theta) = -\nabla_\theta^2 \mathcal{L}(\theta; \mathcal{D}). \tag{55}$$

This shows that contrastive loss minimization is indeed a likelihood maximization. We highlight that this proof assumes margin $m > 2$, i.e. all negatives are inside the margin (because on the manifold $\mathcal{S}^Z$ the maximum distance is 2). Following the exact same structure, a proof can be done with arbitrary $m$. The only difference would be in the definition Eq. 37, instead of a Von Mises-Fisher it would be a discontinuous density and the normalization constants would depend on both $\kappa$ and $m$.

## E   Bayesian Metric learning

Having defined the dataset likelihood $\mathcal{P}(\mathcal{D}|\theta)$ conditioned to the parameter $\theta$, the parameter likelihood $\mathcal{P}(\theta|\mathcal{D})$ conditioned to the dataset is defined according to Bayes rule as

$$\mathcal{P}(\theta|\mathcal{D}) = \frac{\mathcal{P}(\mathcal{D}|\theta)\mathcal{P}(\theta)}{\mathcal{P}(\mathcal{D})}. \tag{56}$$

The aim of metric learning is to maximise such likelihood with respect to the parameter $\theta$, specifically, it aims at finding the Maximum A Posteriori

$$\theta^{\text{MAP}} \in \arg\max_{\theta \in \Theta} \mathcal{P}(\theta|\mathcal{D}). \tag{57}$$

We highlight that the argmax is the same in log-scale

$$\arg\max_{\theta \in \Theta} \mathcal{P}(\theta|\mathcal{D}) = \arg\max_{\theta \in \Theta} \log \mathcal{P}(\theta|\mathcal{D}) = \arg\max_{\theta \in \Theta} \log \mathcal{P}(\mathcal{D}|\theta) + \log \mathcal{P}(\theta) \tag{58}$$

where the log-prior takes the form of a standard $l^2$ regularize (weight decay), and, as shown before, maximizing the log-likelihood $\log p(\mathcal{D}|\theta)$ is equivalent to minimizing of the contrastive loss $\mathcal{L}(\theta; \mathcal{D})$.

However in the Bayesian setting, we do not seek a maximum, but we seek an expression of the whole distribution $\mathcal{P}(\theta|\mathcal{D})$, for which we use the notation $q(\theta)$. We do that by maximizing the dataset likelihood $\mathcal{P}(\mathcal{D})$ by integrating out $\theta$ with $\mathcal{P}(\mathcal{D}) = \mathbb{E}_{\theta \sim q}[\mathcal{P}(\mathcal{D}|\theta)]$. This maximization is then defined by

$$q^*(\theta) := \mathcal{P}^*(\theta|\mathcal{D}) \in \arg\max_{\mathcal{P}(\theta|\mathcal{D}) \in \mathcal{G}(\Theta)} \mathbb{E}_{\theta \sim \mathcal{P}(\theta|\mathcal{D})}[\mathcal{P}(\mathcal{D}|\theta)] \tag{59}$$

Notice that, limited by the ability to parametrize distributions, we aim at finding such a maximum on $q$ over only some subspace of distributions on $\Theta$. To this end, we consider the Laplace approximation, which is one way of choosing a subspace of the parameter distribution. In Laplace post-hoc we restrict ourselves to the space of Gaussians $\mathcal{G}(\Theta) \subset \Delta(\Theta)$ centered in $\theta^{\text{MAP}}$. This is a strong assumption since there are (1) no guarantees of $p(\theta|\mathcal{D})$ being Gaussian and (2) no guarantees of the distribution to be centered in $\theta^{\text{MAP}}$. Online Laplace lifts the latter assumption, and only assumes the parameters to be Gaussian distributed.

With the general Bayesian framework in mind, we now proceed to derive post-hoc and online Laplace.

### E.1 Laplace post-hoc

The Bayes rule

$$P(\theta|\mathcal{D}) = \frac{P(\mathcal{D}|\theta)P(\theta)}{P(\mathcal{D})} \tag{60}$$

implies that

$$\nabla_\theta^2 \log P(\theta|\mathcal{D}) = \nabla_\theta^2 \log P(\mathcal{D}|\theta) + \nabla_\theta^2 \log P(\theta). \tag{61}$$

Assuming an isotropic Gaussian prior $P(\theta) \sim \mathcal{N}(\theta|0, \sigma_{\text{prior}}^2 \mathbb{I})$ implies $\nabla_\theta^2 \log P(\theta) = -\sigma_{\text{prior}}^{-2} \mathbb{I}$ and we have

$$\nabla_\theta^2 \log P(\theta|\mathcal{D}) = \nabla_\theta^2 \log P(\mathcal{D}|\theta) - \sigma_{\text{prior}}^{-2} \mathbb{I} \tag{62}$$

$$= -\nabla_\theta^2 \mathcal{L}(\theta; \mathcal{D}) - \sigma_{\text{prior}}^{-2} \mathbb{I}. \tag{63}$$

Thus, we have two options:

- IF $P(\theta|\mathcal{D})$ is a Gaussian it holds $\forall \theta^* \in \Theta$

$$P(\theta|\mathcal{D}) \sim \mathcal{N}(\theta|\mu = \theta^{\text{MAP}}, \Sigma = (\nabla_\theta^2 \mathcal{L}(\theta^*; \mathcal{D}) + \sigma_{\text{prior}}^{-2} \mathbb{I})^{-1}) \tag{64}$$

- ELSE we can do a second order Taylor approximation of $\log P(\theta|\mathcal{D})$ around $\theta^{\text{MAP}}$ and we have the approximation

$$P(\theta|\mathcal{D}) \sim \mathcal{N}(\theta|\mu = \theta^{\text{MAP}}, \Sigma = (\nabla_\theta^2 \mathcal{L}(\theta^{\text{MAP}}; \mathcal{D}) + \sigma_{\text{prior}}^{-2} \mathbb{I})^{-1}) \tag{65}$$

### E.2 Laplace online

At every step $t$ we have some Gaussian on the parameter space

$$q^t(\theta) \sim \mathcal{N}(\theta|\mu = \theta_t, \Sigma = (H_t)^{-1}) \tag{66}$$

Where the values $\theta_t$ and $H_t$ are iteratively defined as

$$\theta_{t+1} = \theta_t + \lambda \nabla_\theta \mathcal{L}(\theta_t; \mathcal{D}) \qquad \theta_0 = \mu_{\text{prior}} \tag{67}$$

and

$$H_{t+1} = (1 - \alpha)H_t + \nabla_\theta^2 \mathcal{L}(\theta_t; \mathcal{D}) \qquad H_0 = \sigma_{\text{prior}}^{-2} \mathbb{I} \tag{68}$$

For some learning rate $\lambda$ and memory factor $\alpha$. This means that $q^0(\theta)$ is actually the prior distribution, which is updated with the first and second order derivatives of the loss. The updates can be improved by computing the derivatives not only in the single point $\theta_t$, but rather on the expected value with $\theta$ following the distribution $q^t$. This leads to the update rules

$$\theta_{t+1} = \theta_t + \lambda \mathbb{E}_{\theta \sim q^t(\theta)}[\nabla_\theta \mathcal{L}(\theta; \mathcal{D})] \qquad \theta_0 = \mu_{\text{prior}} \tag{69}$$

and

$$H_{t+1} = (1 - \alpha)H_t + \mathbb{E}_{\theta \sim q^t(\theta)}[\nabla_\theta^2 \mathcal{L}(\theta; \mathcal{D})] \qquad H_0 = \sigma_{\text{prior}}^{-2} \mathbb{I} \tag{70}$$

## F Derivatives

In order to perform the Laplace based learning, we need to compute the second-order derivative of the loss with respect to the parameter $\theta$. Let us start by fixing a target $y \in \mathbb{R}$ and two data points $x_1, x_2 \in \mathcal{X}$ and compute the second order derivative of one contrastive term $\mathcal{L}_y(f_\theta(x_1), f_\theta(x_2))$. This term can be viewed as a composition of functions, graphically represented as

$$\begin{array}{ccccc} x_1 & \xrightarrow{\quad f_\theta \quad} & z_1 & \xrightarrow{\quad \mathcal{L}_y \quad} & l \\ x_2 & \xrightarrow[\quad f_\theta \quad]{} & z_2 & & \end{array} \tag{71}$$

To make the derivation cleaner is it useful to define, for a given function $f_\theta : \mathcal{X} \to \mathcal{Z}$, and auxiliary function $\mathfrak{F}_\theta : \mathcal{X}^2 \to \mathcal{Z}^2$ defined by $\mathfrak{F}_\theta(x_1, x_2) := (f_\theta(x_1), f_\theta(x_2))$. In this way the graphical representation is

$$\begin{pmatrix} x_1 \\ x_2 \end{pmatrix} \xrightarrow{\quad \mathfrak{F}_\theta \quad} \begin{pmatrix} z_1 \\ z_2 \end{pmatrix} \xrightarrow{\quad \mathcal{L}_y \quad} l \tag{72}$$

and we can directly apply the chain rule. Before doing so, it is convenient to expand some derivatives to express the $\mathcal{Z}^2$-size matrix as two $\mathcal{Z}$-size submatrixes. The Jacobian of $\mathfrak{F}$ evaluated in $(x_1, x_2)$, is an operator from the tangent space $T_\theta \Theta$ to the tangent space $T_{(z_1, z_2)} \mathcal{Z}^2$, which can be written in block matrix form as

$$J_\theta \mathfrak{F}_\theta(x_1, x_2) = \begin{pmatrix} J_\theta f_\theta(x_1) \\ J_\theta f_\theta(x_2) \end{pmatrix} \tag{73}$$

and, similarly, the Hessian of $\mathcal{L}_y(z_1, z_2)$ can be written in block form as

$$\nabla^2_{(z_1, z_2)} \mathcal{L}_y(z_1, z_2) = \begin{pmatrix} \nabla^2_{z_1} \mathcal{L}_y(z_1, z_2) & \nabla_{z_1} \nabla_{z_2} \mathcal{L}_y(z_1, z_2) \\ \nabla_{z_2} \nabla_{z_1} \mathcal{L}_y(z_1, z_2) & \nabla^2_{z_2} \mathcal{L}_y(z_1, z_2) \end{pmatrix} \tag{74}$$

The Hessian of the per-observation Contrastive loss is then

$$\nabla^2_\theta \mathcal{L}_y(f_\theta(x_1), f_\theta(x_2)) = \nabla^2_\theta \mathcal{L}_y(\mathfrak{F}_\theta(x_1, x_2)) \tag{75}$$

$$\overset{\text{GGN}}{\approx} J_\theta \mathfrak{F}_\theta(x_1, x_2)^\top \cdot \nabla^2_{(z_1, z_2)} \mathcal{L}_y(z_1, z_2) \cdot J_\theta \mathfrak{F}_\theta(x_1, x_2) \tag{76}$$

$$= \begin{pmatrix} J_\theta f_\theta(x_1) \\ J_\theta f_\theta(x_2) \end{pmatrix}^\top \begin{pmatrix} \nabla^2_{z_1} \mathcal{L}_y(z_1, z_2) & \nabla_{z_1} \nabla_{z_2} \mathcal{L}_y(z_1, z_2) \\ \nabla_{z_2} \nabla_{z_1} \mathcal{L}_y(z_1, z_2) & \nabla^2_{z_2} \mathcal{L}_y(z_1, z_2) \end{pmatrix} \begin{pmatrix} J_\theta f_\theta(x_1) \\ J_\theta f_\theta(x_2) \end{pmatrix} \tag{77}$$

where $(z_1, z_2) = h_\theta(x_1, x_2)$ is the point where we have to evaluate the derivative wrt to $z$. Consequently, the Hessian of the contrastive loss is

$$\nabla^2_\theta \mathcal{L}(\theta; \mathcal{D}) = \sum_{\mathfrak{p}_{ij} \in \mathcal{D}^2} \nabla^2_\theta \mathcal{L}_{y_{ij}}(f_\theta(x_i), f_\theta(x_j)) \tag{78}$$

$$= \sum_{\mathfrak{p}_{ij} \in \mathcal{D}^2} \begin{pmatrix} J_\theta f_\theta(x_i) \\ J_\theta f_\theta(x_j) \end{pmatrix}^\top \begin{pmatrix} \nabla^2_{z_i} \mathcal{L}_{y_{ij}}(z_i, z_j) & \nabla_{z_j} \nabla_{z_j} \mathcal{L}_{y_{ij}}(z_i, z_j) \\ \nabla_{z_j} \nabla_{z_i} \mathcal{L}_{y_{ij}}(z_i, z_j) & \nabla^2_{z_j} \mathcal{L}_{y_{ij}}(z_i, z_j) \end{pmatrix} \begin{pmatrix} J_\theta f_\theta(x_i) \\ J_\theta f_\theta(x_j) \end{pmatrix} \tag{79}$$

We now proceed to find the derivatives of the per-observation loss $\mathcal{L}_y$ wrt. the Neural Network outputs $z_i = f_\theta(x_i)$ and $z_j = f_\theta(x_j)$. The derivatives varies based on the specific loss term that we consider. In the following we derive the derivatives for Euclidean $\mathcal{L}_y^E$ and Arccos $\mathcal{L}_y^A$ cases.

**Split choice**. The GGN assumes access to a composition of two functions, the specific split choice affects the result. As said in Appendix C it is common to have a normalization layer at the end of the Neural Network. This can be schematized as follow

$$\begin{pmatrix} x_1 \\ x_2 \end{pmatrix} \xrightarrow{\text{NN}} \begin{pmatrix} z_1 \\ z_2 \end{pmatrix} \xrightarrow{\ell_2\text{-norm}} \begin{pmatrix} \frac{z_1}{\|z_1\|} \\ \frac{z_2}{\|z_2\|} \end{pmatrix} \xrightarrow{\text{distance}} l \tag{80}$$

and this leads to (at least) two possible split choices, wheter we include the normalization layer in the network left or right function, which we proceed to study further. We highlight that these two split choices can be interpreted as different loss function or, equivalently, as different distance metric: Euclidean or Arccos.

## F.1 Euclidean distance

If we consider the $\ell_2$-normalization layer as part of the Neural Network $f$

$$\begin{pmatrix} x_1 \\ x_2 \end{pmatrix} \xrightarrow{\text{NN}} \begin{pmatrix} z_1 \\ z_2 \end{pmatrix} \xrightarrow{\ell_2\text{-norm}} \underbrace{\begin{pmatrix} \frac{z_1}{\|z_1\|} \\ \frac{z_2}{\|z_2\|} \end{pmatrix} \xrightarrow{\text{distance}} l}_{\mathcal{L}_y^E} \tag{81}$$

then the loss $\mathcal{L}_y = \mathcal{L}_y^E = $ is the *Euclidean* distance defined as

$$\mathcal{L}_y^E(z_i, z_j) := \frac{1}{2} y \|z_i - z_j\|^2 \tag{82}$$

and it holds

$$\nabla^2_{z_1}\mathcal{L}^E_y(z_1, z_2) = \nabla^2_{z_2}\mathcal{L}^E_y(z_1, z_2) = y\mathbb{I} \tag{83}$$

$$\nabla_{z_1}\nabla_{z_2}\mathcal{L}^E_y(z_1, z_2) = \nabla_{z_2}\nabla_{z_1}\mathcal{L}^E_y(z_1, z_2) = -y\mathbb{I} \tag{84}$$

which leads to

$$\nabla^2_{(z_1,z_2)}\mathcal{L}^E_y(z_1, z_2) = y\begin{pmatrix} \mathbb{I} & -\mathbb{I} \\ -\mathbb{I} & \mathbb{I} \end{pmatrix}. \tag{85}$$

We highlight that the matrix $\begin{pmatrix} \mathbb{I} & -\mathbb{I} \\ -\mathbb{I} & \mathbb{I} \end{pmatrix}$ is positive semi-definite, this will be useful in Proposition F.3.

**Intuition.** The overall Hessian expression can be further simplified by considering the matrix square root.

$$\nabla^2_\theta\mathcal{L}(\theta; \mathcal{D}) = \sum_{\mathfrak{p}_{ij}\in\mathcal{D}^2} \nabla^2_\theta\mathcal{L}_{y(p_{ij})}(f_\theta(x_i), f_\theta(x_j)) \tag{86}$$

$$= \sum_{\mathfrak{p}_{ij}\in\mathcal{D}^2} y_{ij} \begin{pmatrix} J_\theta f_\theta(x_i) \\ J_\theta f_\theta(x_j) \end{pmatrix}^\top \begin{pmatrix} \mathbb{I} & -\mathbb{I} \\ -\mathbb{I} & \mathbb{I} \end{pmatrix} \begin{pmatrix} J_\theta f_\theta(x_i) \\ J_\theta f_\theta(x_j) \end{pmatrix} \tag{87}$$

$$= \frac{1}{2} \sum_{\mathfrak{p}_{ij}\in\mathcal{D}^2} y_{ij} \begin{pmatrix} J_\theta f_\theta(x_i) \\ J_\theta f_\theta(x_j) \end{pmatrix}^\top \begin{pmatrix} \mathbb{I} & -\mathbb{I} \\ -\mathbb{I} & \mathbb{I} \end{pmatrix} \begin{pmatrix} \mathbb{I} & -\mathbb{I} \\ -\mathbb{I} & \mathbb{I} \end{pmatrix} \begin{pmatrix} J_\theta f_\theta(x_i) \\ J_\theta f_\theta(x_j) \end{pmatrix} \tag{88}$$

$$= \frac{1}{2} \sum_{\mathfrak{p}_{ij}\in\mathcal{D}^2} y_{ij} \begin{pmatrix} J_\theta f_\theta(x_i) - J_\theta f_\theta(x_j) \\ J_\theta f_\theta(x_j) - J_\theta f_\theta(x_i) \end{pmatrix}^\top \begin{pmatrix} J_\theta f_\theta(x_i) - J_\theta f_\theta(x_j) \\ J_\theta f_\theta(x_j) - J_\theta f_\theta(x_i) \end{pmatrix} \tag{89}$$

$$= \sum_{\mathfrak{p}_{ij}\in\mathcal{D}^2} y_{ij} \left( J_\theta f_\theta(x_i) - J_\theta f_\theta(x_j) \right)^\top \left( J_\theta f_\theta(x_i) - J_\theta f_\theta(x_j) \right). \tag{90}$$

This expression give raise to two interpretations. (1) This formulation draws parallels with the GGN approximation of the MSE loss. (2) It can be viewed as the squared distance of the jacobians product, where the sign $y_{ij}$ determines the sign. This is parallel with the contrastive loss

$$\mathcal{L}(\theta; \mathcal{D}) = \frac{1}{2} \sum_{\mathfrak{p}_{ij}\in\mathcal{D}^2} y_{ij}\|f_\theta(x_i) - f_\theta(x_j)\|^2 \tag{91}$$

$$= \frac{1}{2} \sum_{\mathfrak{p}_{ij}\in\mathcal{D}^2} y_{ij} \left( f_\theta(x_i) - f_\theta(x_j) \right)^\top \left( f_\theta(x_i) - f_\theta(x_j) \right) \tag{92}$$

with the only difference being the Jacobian operator (and the 2 factor).

### F.2 Arccos distance

If we consider the $\ell_2$-normalization layer as part of the loss function $\mathcal{L}_y$

$$\begin{pmatrix} x_1 \\ x_2 \end{pmatrix} \xrightarrow{\text{NN}} \begin{pmatrix} z_1 \\ z_2 \end{pmatrix} \underbrace{\xrightarrow{\ell_2\text{-norm}} \begin{pmatrix} \frac{z_1}{\|z_1\|} \\ \frac{z_2}{\|z_2\|} \end{pmatrix} \xrightarrow{\text{distance}} l}_{\mathcal{L}^A_y} \tag{93}$$

then the loss $\mathcal{L}_y = \mathcal{L}^A_y$ is the *Arccos* distance defined as

$$\mathcal{L}^A_y(z_i, z_j) := \frac{1}{2}y\left\|\frac{z_i}{\|z_i\|} - \frac{z_j}{\|z_j\|}\right\|^2 = y\left(1 - \left\langle \frac{z_i}{\|z_i\|}, \frac{z_j}{\|z_j\|} \right\rangle\right) \tag{94}$$

and it holds

$$\nabla_{z_1}^2 \mathcal{L}_y^A(z_1, z_2) = \frac{y}{\|z_1\|^2}\left(\left\langle \frac{z_1}{\|z_1\|}, \frac{z_2}{\|z_2\|}\right\rangle \mathbb{I} + \frac{z_1^\top z_2 + z_2^\top z_1}{\|z_1\|\|z_2\|} - 3\left\langle \frac{z_1}{\|z_1\|}, \frac{z_2}{\|z_2\|}\right\rangle \frac{z_1^\top z_1}{\|z_1\|^2}\right) \tag{95}$$

$$\nabla_{z_2}\nabla_{z_1} \mathcal{L}_y^A(z_1, z_2) = \frac{y}{\|z_1\|\|z_2\|}\left(-\mathbb{I} + \frac{z_1^\top z_1}{\|z_1\|^2} + \frac{z_2^\top z_2}{\|z_2\|^2} - \left\langle \frac{z_1}{\|z_1\|}, \frac{z_2}{\|z_2\|}\right\rangle \frac{z_1^\top z_2}{\|z_1\|\|z_2\|}\right) \tag{96}$$

$$\nabla_{z_2}^2 \mathcal{L}_y^A(z_1, z_2) = \frac{y}{\|z_2\|^2}\left(\left\langle \frac{z_1}{\|z_1\|}, \frac{z_2}{\|z_2\|}\right\rangle \mathbb{I} + \frac{z_1^\top z_2 + z_2^\top z_1}{\|z_1\|\|z_2\|} - 3\left\langle \frac{z_1}{\|z_1\|}, \frac{z_2}{\|z_2\|}\right\rangle \frac{z_2^\top z_2}{\|z_2\|^2}\right) \tag{97}$$

### F.3 Hessian approximations

In classification and regression tasks, the Generalized Gauss Newton approximation is sufficient to guarantee positive definiteness, this is because the Hessian of the loss with respect to the Neural Network output $\nabla_z^2 \mathcal{L}_y(z)$ is positive definite both for MSE and cross-entropy. The overall loss in those cases is a sum over data points, without subtraction, and thus a positive matrix. For the contrastive loss, on the other hand, positive pairs contribute positively while negative pairs contribute negatively, and thus there is no guarantee in general. Moreover, with the Arccos loss there is not even the guarantee that $\nabla_z^2 \mathcal{L}_y^A(z)$ is positive definite. Therefore, we consider three approximations of the Hessian of the contrastive loss: full, positives, fixed, that ensures it to positive definite.

**Full.** The first possible approach is to forcefully ensure that the matrix only has positive values. This is formalized by applying an elementwise ReLU, that is

$$\left[\nabla_\theta^2 \mathcal{L}(\theta; \mathcal{D})\right]_{nm} \approx \max\left(0, \left[\nabla_\theta^2 \mathcal{L}(\theta; \mathcal{D})\right]_{nm}\right) \qquad \forall n, m \tag{98}$$

**Positives.** An alternative approach is to consider only the contribution of the positive pairs

$$\nabla_\theta^2 \mathcal{L}(\theta; \mathcal{D}) \approx \sum_{\mathfrak{p}_{ij} \in \mathcal{D}_{\text{pos}}^2} \nabla_\theta^2 \mathcal{L}_{y_{ij}}(f_\theta(x_i), f_\theta(x_j)) \tag{99}$$

neglecting the contribution of the negative pairs $\sum_{\mathcal{D}_{\text{neg}}^2}$. This approximation will be far from the truth and there is no theoretical justification. This approximation strategy is inspired by Shi and Jain [8], which only uses positive pairs to train the network responsible for predicting the variance.

**Fixed.** The third approach is to consider the contrastive term as a function of *one data point at a time*, assuming the other fixed. This idea can be formalized by making use of the stop gradient notation: sg. The per-observation contrastive Eq. 12 can be written as

$$\mathcal{L}_y(z_1, z_2) = \frac{1}{2}\mathcal{L}_y(\text{sg}[z_1], z_2) + \frac{1}{2}\mathcal{L}_y(z_1, \text{sg}[z_2]). \tag{100}$$

We highlight that, with this definition, the zero- and first-order derivative does not change, and thus the loss and gradient are exactly the same as for the standard contrastive loss. However, the second-order derivative looses the cross term

$$\nabla_{(z_1, z_2)}^2 \mathcal{L}_y(z_1, z_2) = \begin{pmatrix} \nabla_{z_1}^2 \mathcal{L}_y(z_1, z_2) & 0 \\ 0 & \nabla_{z_2}^2 \mathcal{L}_y(z_1, z_2) \end{pmatrix} \tag{101}$$

and the Hessian of the loss can then be more compactly written as

$$\begin{aligned}
\nabla_\theta^2 \mathcal{L}(\theta; \mathcal{D}) &= \sum_{\mathfrak{p}_{ij} \in \mathcal{D}^2} \nabla_\theta^2 \mathcal{L}_{y_{ij}}(f_\theta(x_i), f_\theta(x_j)) \\
&= \sum_{\mathfrak{p}_{ij} \in \mathcal{D}^2} y_{ij} \begin{pmatrix} J_\theta f_\theta(x_i) \\ J_\theta f_\theta(x_j) \end{pmatrix}^\top \begin{pmatrix} \nabla_{z_i}^2 \mathcal{L}_{y_{ij}}(z_i, z_j) & 0 \\ 0 & \nabla_{z_j}^2 \mathcal{L}_{y_{ij}}(z_i, z_j) \end{pmatrix} \begin{pmatrix} J_\theta f_\theta(x_i) \\ J_\theta f_\theta(x_j) \end{pmatrix} \\
&= 2\sum_{\mathfrak{p}_{ij} \in \mathcal{D}^2} y_{ij}\, J_\theta f_\theta(x_i)^\top \nabla_{z_i}^2 \mathcal{L}_{y_{ij}}(z_i, z_j)\, J_\theta f_\theta(x_i). 
\end{aligned} \tag{102}$$

**Proposition F.1.** *Consider the **full** approximation. Using only the diagonal of the Hessian $\implies$ positive definiteness.*

*Proof.* The eigenvalues of a diagonal matrix are exactly the values on the diagonal. Enforcing these elements to be positive is equivalent to enforcing that all eigenvalues are positive. This implies the positive definiteness of the matrix. □

**Proposition F.2.** *Consider the **positives** approximation. Euclidean loss $\implies$ positive definiteness.*

*Proof.* Consider the Hessian of Eq. 99 and substitute the expression Eq. 85

$$\nabla_\theta^2 \mathcal{L}(\theta; \mathcal{D}) \approx \sum_{\mathfrak{p}_{ij} \in \mathcal{D}_{\text{pos}}^2} \nabla_\theta^2 \mathcal{L}_{y_{ij}}(f_\theta(x_i), f_\theta(x_j)) \tag{103}$$

$$= \sum_{\mathfrak{p}_{ij} \in \mathcal{D}_{\text{pos}}^2} y_{ij} \begin{pmatrix} J_\theta f_\theta(x_i) \\ J_\theta f_\theta(x_j) \end{pmatrix}^\top \begin{pmatrix} \mathbb{I} & -\mathbb{I} \\ -\mathbb{I} & \mathbb{I} \end{pmatrix} \begin{pmatrix} J_\theta f_\theta(x_i) \\ J_\theta f_\theta(x_j) \end{pmatrix} \tag{104}$$

$$= \frac{1}{|\mathcal{D}_{\text{pos}}^2|} \sum_{\mathfrak{p}_{ij} \in \mathcal{D}_{\text{pos}}^2} \begin{pmatrix} J_\theta f_\theta(x_i) \\ J_\theta f_\theta(x_j) \end{pmatrix}^\top \begin{pmatrix} \mathbb{I} & -\mathbb{I} \\ -\mathbb{I} & \mathbb{I} \end{pmatrix} \begin{pmatrix} J_\theta f_\theta(x_i) \\ J_\theta f_\theta(x_j) \end{pmatrix} \tag{105}$$

which is a sum of positive definite matrixes. □

**Proposition F.3.** *Consider the **fixed** approximation. Euclidean loss $\implies$ positive definiteness.*

*Proof.* Consider the Hessian of Eq. 102 and substitute the expression Eq. 85

$$\nabla_\theta^2 \mathcal{L}(\theta; \mathcal{D}) = 2 \sum_{\mathfrak{p}_{ij} \in \mathcal{D}^2} y_{ij} \, J_\theta f_\theta(x_i)^\top J_\theta f_\theta(x_i) \tag{106}$$

$$= 2 \sum_{x_i, c_i \in \mathcal{D}} \sum_{x_j, c_j \in \mathcal{D}} y_{ij} \, J_\theta f_\theta(x_i)^\top J_\theta f_\theta(x_i) \tag{107}$$

$$= 2 \sum_{x_i, c_i \in \mathcal{D}} \left( \sum_{x_j, c_j \in \mathcal{D}} y_{ij} \right) J_\theta f_\theta(x_i)^\top J_\theta f_\theta(x_i) \tag{108}$$

$$= 2 \sum_{x_i, c_i \in \mathcal{D}} \left( \frac{|\mathcal{D}_{\text{pos}}^2(x_i)|}{|\mathcal{D}_{\text{pos}}^2(x_i)|} - \frac{|\mathcal{D}_{\text{neg inside}}^2(x_i)|}{|\mathcal{D}_{\text{neg}}^2(x_i)|} \right) J_\theta f_\theta(x_i)^\top J_\theta f_\theta(x_i) \tag{109}$$

$$= 2 \sum_{x_i, c_i \in \mathcal{D}} \underbrace{\left( 1 - \frac{|\mathcal{D}_{\text{neg inside}}^2(x_i)|}{|\mathcal{D}_{\text{neg}}^2(x_i)|} \right)}_{\geq 0} J_\theta f_\theta(x_i)^\top J_\theta f_\theta(x_i) \tag{110}$$

where $|\mathcal{D}_{\text{pos}}^2(x_i)|$ is the number of positive pairs containing the point $x_i$, and similarly for $|\mathcal{D}_{\text{neg}}^2(x_i)|$ and $|\mathcal{D}_{\text{neg inside}}^2(x_i)|$. The Hessian is then equivalent to a sum of positive definite matrixes multiplied by a non-negative factor, and thus it is positive definite. □

We highlight, as we can see from the last Proposition's proof, that if all the negatives are inside the margin, the Hessian is exactly 0. The more negatives are outside the margin, the more positive the Hessian is.

# G   Experimental details

In this section, we provide details on the experiments. We highlight that code for all experiments and baselines are available at `https://github.com/****/bayesian-metric-learning`. (supplementary)

### G.1 Datasets

**FashionMnist.** We use the standard test-train split for FashionMnist [15] and for MNIST [5]. We normalize the images in the range $[0, 1]$ and do not perform any data augmentation during training. We train with RMSProp with learning rate $10^{-3}$ and default PyTorch settings, and with exponential learning rate decay with $\gamma = \exp(-0.1)$. We use a memory factor of 0.0001 and maximum 5000 pairs per train step. We trained for 20 epochs.

**CIFAR10.** We use the standard test-train split for CIFAR [] and for SVHN []. We normalize the images in the range $[0, 1]$ and do not perform any data augmentation during training. We train with RMSProp with learning rate $10^{-5}$ and default PyTorch settings, and with exponential learning rate decay with $\gamma = \exp(-0.1)$. We use a memory factor of 0.0001 and maximum 5000 pairs per train step. We trained for 20 epochs.

**CUB200.** The CUB-200-2011 dataset [11] consists of $11\,788$ images of 200 bird species. The birds are captured from different perspectives and in different environments, making this a challenging dataset for image retrieval. We follow the procedure of Musgrave et al. [7] and divide the first 100 classes into the training set and the last 100 classes into the test set. In this zero-shot setting, the trained models have not seen any of the bird species in the test set, and the learned features must generalize well across species. Similarly to Warburg et al. [14], we use the Stanford Car-196 [4] dataset as OoD data. The Car-196 dataset is composed of $16\,185$ images of 196 classes of cars. We conduct a similar split as Musgrave et al. [7], and only evaluate on the last 98 classes. This constitutes a very challenging zero-shot OoD dataset, where the model at test time needs to distinguish cars from birds, however, the model has not seen any of the bird species at training time. We use imagenet normalization and during training augment with random resized crops and random horizontal flipping. We image $224x224$ image resolution. We train with RMSProp with learning rate $10^{-7}$ and default PyTorch settings, and with exponential learning rate decay with $\gamma = \exp(-0.1)$. We use a memory factor of 0.0001 and maximum 30 pairs per train step. We trained for 20 epochs.

**LFW.** We use the face recognition dataset LFW [2] with the standard zero-shot train/test split, CUB200 as OoD data. It is challenging because of the high number of classes and few observations per class. LFW [2] consists of $13\,000$ images of $5\,749$ people using the standard zero-shot train/test split. One reason reliable uncertainties are important for face recognition systems is to avoid granting access based on an erroneous prediction. An example of such failure happened with the initial release of the Apple Face ID software, which failed to recognize underrepresented groups that were missing or underrepresented in the training distribution [10]. Reliable uncertainties and OoD detection might have mitigated such issues. We use imagenet normalization and during training augment with random resized crops and random horizontal flipping. We image $224x224$ image resolution. We train with RMSProp with learning rate $10^{-7}$ and default PyTorch settings, and with exponential learning rate decay with $\gamma = \exp(-0.1)$. We use a memory factor of 0.0001 maximum 30 pairs per train step. We trained for 200 epochs.

**MSLS.** We use standard zero-shot train/val/test split [13]. We image $224x224$ image resolution. We train with RMSProp with learning rate $10^{-7}$ and default PyTorch settings, and with exponential learning rate decay with $\gamma = \exp(-0.1)$. We use a memory factor of 0.0001 maximum 10 pairs per train step. We trained for 200 epochs.

### G.2 Model Architectures

For all experiments, we use a last-layer diagonal LA. Across all experiments, our networks follow standard practices in image retrieval. For FashionMnist, the network is [`conv2d(1,32)`, `relu, conv2d(32, 64), relu, maxpool2d(2), Flatten, Linear(9216)`] and for CiFAR10, the network is [`conv2d(1,32), relu, conv2d(32, 64), relu, maxpool2d(2), conv2d(64,64), relu, Flatten, Linear(9216)`]. For CUB200 and LFW, we use a pretrained ResNet50 backbone followed by a Generalized-Mean pooling layer and a dimension-preserving learned whitening layer (linear layer). The weight posterior is learned only for the last layer

### G.3 Metrics

Evaluating the models' uncertainty estimates is tricky. We propose four metrics that capture both *interpolation* and *extrapolation* behaviors uncertainty estimates. To measure *extrapolation* behavior, we measure the performance of out-of-distribution (OOD) detection and report Area Under Receiver Operator Curve (AUROC) and Area Under Precision-Recall Curve (AUPRC). These metrics describe the model's ability to assign high uncertainty to observations it has not seen during training (e.g., we train a model on birds and use images of cars as out-of-distribution examples), and are often used in unsupervised representation learning [6]. To measure *interpolation* behavior, we measure the models' ability to assign reliable uncertainties to in-distribution data. Similarly to Wang and Deng [12], we measure the Area Under the Sparsification Curve. The sparsification curve is computed by iteratively removing the observation with the highest variance and recomputing the predictive performance (mAP@1). Lastly, for closed set datasets, we propose a method to compute Expected Calibration Error (ECE) for image retrieval. This is done by sampling $100$ latent variables $z_i$ from the predicted latent distributions $p(z|I)$. We then perform nearest neighbor classification on these samples to obtain $c_i$ predictions. We use the mode as the model prediction and the consensus with mode as the confidence, e.g., if they constitute $60$ samples, then the prediction has $60\%$ confidence. We compare the estimated confidence with accuracy with

$$\text{ECE} = \sum_{i}^{N} \frac{1}{|B_i|} \left| \text{acc}(B_i) - \text{conf}(B_i) \right|, \tag{111}$$

where we $\text{acc}(B_i)$ and $\text{conf}(B_i)$ is the accuracy and confidence of the $i^{\text{th}}$ bin.

### G.4 Details on baselines

If nothing else is stated we use the same architecture and training/evaluation code for all models. We also release all baseline models.

- Deterministic: we train with the contrastive loss.
- MC dropout: we train with dropout between all trainable layers (except for the ResNet50 blocks). We use a dropout rate $0.2$. Dropout was enabled during test time.
- Deep Ensemble: Each ensemble consists of $5$ models, each initialized with different seeds.
- PFE: We add an uncertainty module that learns the variance. The rest of the model is frozen during training and initialized with a deterministically trained model. The uncertainty model consists of [linear, batchnorm1d, relu, linear, batchnorm1d]. We experimented with the last batch normalization (BN) layer sharing parameters or just a standard BN layer. We found the former to work slightly better.
- HIB: We use the same uncertainty module as in PFE, and a kl weight of $10^{-4}$. We use $8$ samples during training to compute the loss.