# OpenReview forum: "Bayesian Metric Learning for Uncertainty Quantification in Image Retrieval"
_NeurIPS.cc/2023/Conference — NeurIPS 2023 poster_

### Official Review · Reviewer_Stu9 · 2023-07-04

**Soundness:** 3 good
**Presentation:** 2 fair
**Contribution:** 3 good
**Rating:** 6
**Confidence:** 4

**Summary:**

This paper presents a Laplace approximation-based probabilistic retrieval approach (aka. Bayesian metric learning for image retrieval). The author provides a probabilistic view of the contrastive loss based on the von-Mises Fisher distribution and corrections for the Hessian positive definiteness. Extensive experimental evaluations are performed, demonstrating the advantages of the presented approach in calibrated uncertainties, out-of-distribution detection, close-set and open-set retrieval, and ablations on a few parameters of design choices.

**Strengths:**

[Originality] This paper presents the use of Laplace approximation instead of amortized inference for probabilistic retrieval. Although similar ideas have been proposed to improve the amortization gap of variational autoencoders in several prior works, its development in the context of metric learning is novel. This includes the introduction of a probabilistic view of the contrastive loss and the utilization of the Hessian approximation based on GCN.

On the other hand, previous Bayesian metric learning approaches have also been proposed, such as hedge instance embedding (HIE) and probabilistic face embeddings (PFE). In this regard, the novelty of this work lies in the contrastive learning-based loss and the learning of stochastic embeddings based on the Laplacian autoencoder (Miani, M. et al., 2022).

Miani, M., Warburg, F., Moreno-Muñoz, P., Skafte, N., & Hauberg, S. (2022). Laplacian autoencoders for learning stochastic representations. *Advances in Neural Information Processing Systems*, *35*, 21059-21072.

[Quality] Extensive empirical evaluations, including a careful ablation study, are conducted. The results demonstrate that the proposed approach outperforms HIE and PFE in the considered cases.

[Clarity] The paper effectively uses figures to demonstrate ideas. However, I do find the need of reading with jumps between the main text and the supplemental material in order to grasp the concept. The organization can be improved.

[Significance] This paper demonstrates uncertainty quantification based on Laplace approximation in the context of probabilistic retrieval. This is an important topic for improving the robustness and mitigating the silent failure of deep neural network systems. The main contribution of this paper appears to be the empirical study, which can be informative and useful for practitioners.

**Weaknesses:**

It appears that this paper builds upon prior work, including the Laplacian autoencoder (Miani, M. et al., 2022), as well as several works on uncertainty in metric learning. Consequently, I am more concerned about the unique technical contributions of this work. In this regard, the necessity of the probabilistic view of the contrastive loss is not well presented and not well motivated.

 Another missing piece is how to perform test-time retrieval within the proposed framework and how its efficiency compares to the amortized approach. Given a query image, does the proposed approach learn a stochastic representation for it? Is the retrieval based on ranking the deterministic similarities between the query image and all other candidate images? If not, how is it performed?

The organization and clarity of this paper could be improved. If certain results are deemed important enough, they should be moved into the main text. The authors should focus on introducing the unique aspects of this work and provide a strong motivation for them, such as the Von Mises-Fisher distribution. The frequent references to the Appendix disrupt the logical flow of the paper.

**Questions:**

The probabilistic view in the supplemental also seems a little bit weird to me since both argument and parameter are data-dependent. Would it be sufficient to just treat the probabilistic contrastive likelihood as a second-order differentiable loss, if they are mathematically equivalent anyway?

What are the main roadblocks that have been overcome in applying the Laplacian autoencoder (LAE) to probabilistic retrieval?

Are there any other ways of approximating Hessian for contrastive loss, while still maintains scalability and positive definiteness? Is KFAC applicable, e.g., Ritter et al., 2018?

Ritter, H., Botev, A., & Barber, D. (2018, January). A scalable laplace approximation for neural networks. In 6th International Conference on Learning Representations, ICLR 2018-Conference Track Proceedings (Vol. 6). International Conference on Representation Learning.

**Limitations:**

The authors discussed limitations about computation load which I think is reasonable.

---

> ### Author Rebuttal · Authors · 2023-08-09
>
> > It appears that this paper builds upon prior work, including the Laplacian autoencoder (Miani, M. et al., 2022), as well as several works on uncertainty in metric learning. Consequently, I am more concerned about the unique technical contributions of this work. In this regard, the necessity of the probabilistic view of the contrastive loss is not well presented and not well motivated.
>
> The technical contribution: The paper proposes to use Laplace for metric learning. This requires (1) proving that the contrastive loss is a log-likelihood because this is a fundamental assumption of the Laplace approximation. This provides the probabilistic motivation that the reviewer request. (2) Identify approximations of the second-order derivative of the loss that ensures that it is semi-positive definite because otherwise, one might end up with a covariance matrix with negative variances. (3) A novel decomposition of the GGN approximation, and (4) “putting things together”, which is all too often trivialized.
>
> > Another missing piece is how to perform test-time retrieval within the proposed framework and how its efficiency compares to the amortized approach. Given a query image, does the proposed approach learn a stochastic representation for it? Is the retrieval based on ranking the deterministic similarities between the query image and all other candidate images? If not, how is it performed?
>
> The model predicts stochastic representations. The test time retrieval is performed by finding the nearest neighbors using the mean of the representation. This gives a ranking and can be computed as efficiently as in the deterministic case. The uncertainty of the retrieved candidate is based on the variances (or concentrations for vMF distributions) of the query and the candidate. Thus, the retrieval system provides a ranking of the nearest neighbors and the uncertainty of each neighbor.
>
> > The organization and clarity of this paper could be improved. If certain results are deemed important enough, they should be moved into the main text. The authors should focus on introducing the unique aspects of this work and provide a strong motivation for them, such as the Von Mises-Fisher distribution. The frequent references to the Appendix disrupt the logical flow of the paper.
>
> Clarity of the presentation: We thank the reviewer for highlighting that the frequent references to the Appendix disrupt the reading flow. We will update the manuscript, such that it can easily be read without referencing the appendix, e.g. move vMF motivation to the main text.
>
> > Questions:
> > The probabilistic view in the supplemental also seems a little bit weird to me since both argument and parameter are data-dependent. Would it be sufficient to just treat the probabilistic contrastive likelihood as a second-order differentiable loss, if they are mathematically equivalent anyway?
>
> The classic contrastive loss is a second-order differentiable loss, but this is *not* sufficient to treat it as a log-likelihood. Sufficient conditions would be to prove positiveness and integrability, specifically:
>
> (1) $loss>0$ everywhere, which is equivalent to $probability=exp(-loss)<1$
>
> (2) $\int exp(-loss) = C < \infty$
>
> Condition (1) is feasible to prove directly. Condition (2) is more tricky, proving the existence of such constant C is feasible, finding the exact value of such constant is more annoying.
> Importantly, such constant C is in general different from 1 (this can be seen for example by considering trivial cases) and thus the actual probabilistic loss needs to be renormalized. This means that in any case we need to introduce a new loss
>
> $$ probabilistic loss = constrastive loss - log(C) $$
>
> This is of course “equivalent” as losses, since we are simply adding a constant term that will be neglected by the gradient.
>
> This approach (modifying the contrastive loss until it is a log-likelihood) is totally ok, and the reviewer may consider it simpler. We argue that our approach (defining a log-likelihood and then showing it is equivalent) is more elegant, and perhaps more fundamental. Building a probabilistic loss from the basics, i.e. repulsive and attractive terms for each pair, leads to a better intuition on what is going on and how the different pairs interact with each other. Moreover, it gives cleaner access to the explicit value of the normalization constant, as an explicit function of the von-mises-fisher normalization constants.
>
> > What are the main roadblocks that have been overcome in applying the Laplacian autoencoder (LAE) to probabilistic retrieval?
>
> The main roadblocks are (1) showing that the contrastive loss is a log-likelihood, (2) prosing novel approximations to ensure that the second-order derivative is semi-positive definite, (3) proposing a novel decomposition of the Hessian, such that the l2 normalization layer is not linearized.
>
> > Are there any other ways of approximating Hessian for contrastive loss, while still maintains scalability and positive definiteness? Is KFAC applicable, e.g., Ritter et al., 2018?
>
> Our methods will work out of the box for KFAC. Although, using ReLU to ensure semi-positive definiteness is only sufficient for the diagonal case. Both fixed and pos approximations can be used.

---

> > ### Comment · Reviewer_Stu9 · 2023-08-19
> >
> > Thank you for providing the rebuttal! I've read it and I maintain my current rating. If "(1) proving that the contrastive loss is a log-likelihood" is viewed as the key technical contribution by the authors, it deserves more presence in the main text - It was put in Appendix D only.

---

> > > ### Author Response · Authors · 2023-08-20
> > > **Thanks**
> > >
> > > Thanks for the support and feedback. To bring more attention to the likelihood-contribution of the paper, we propose to include a proof sketch showing that the contrastive loss is indeed a log-likelihood.

---

### Official Review · Reviewer_Z8Se · 2023-07-05

**Soundness:** 4 excellent
**Presentation:** 3 good
**Contribution:** 4 excellent
**Rating:** 7
**Confidence:** 5

**Summary:**


The authors show that contrastive loss can be viewed as a likelihood after projection onto the spherical space. This consequently allows them to use the Laplace approximation to estimate the posterior over the parameters. To make the construction further amenable to estimation, the authors propose approaches to make the Hessian positive definite. Finally, the experimental results and ablations go into details about the design choices.

**Strengths:**

- The authors use a conceptually simple approach to constructing approximate posteriors on top of neural networks using the Laplace approximation.
- The authors provide several analogies for various concepts introduced which makes the reader comfortable, and would be resourceful paper for the community.
- Most design choices in the method are covered by ablations, which is much appreciated.

**Weaknesses:**

- The Laplace approximation hinges on an assumption of vanishing gradient which may not be true for modern large neural networks and large datasets used for representation learning. See Question 2.
- The overall method involves a fair number of moving parts, and it would be good to reconcile them as a single algorithm or a list of bullet points for easy digestion for the reader.
- The method can be computationally expensive, and the authors have to resort to construct stochasticity of the last-layer parameters only. Much of the post-hoc LA literature seems to be working with a similar setup, so I do not count this as a major weakness of this paper but rather the whole community.

### Minor

- Please use `\citet` instead of `\citep` for references directly referring to the paper, for instance in Line 164.
- It would be great to have Figure 3 on the same page as the text on ensuring positive definiteness on Page 4.

**Questions:**

1. In Line 46, the authors claim that they do not assume any distribution on the stochastic embeddings. But the choice of using Laplacian approximation on the parameters to construct the posterior implicitly makes an assumption on the constructed embeddings. Could the authors clarify this?
2. As stated in Eq. (2), the Laplace approximation comes from a second-order Taylor expansion, where the first-order gradient vanishes due to $\theta^\star$ being the optima. Does this does really happen in practice, and how numerically close are we to zero for instance w.r.t. the size of models/data?
3. Could the authors confirm if my understanding of the overall approach is correct? (a) Using online LA to construct the posterior (b) Use samples from the posterior to construct stochastic embeddings (c) Use embedding samples to construct $\kappa$ for the von Mises-Fisher distribution which is used as a single number to quantify uncertainty.

**Limitations:**

Yes. See also weaknesses and questions.

---

> ### Author Rebuttal · Authors · 2023-08-09
>
> > The overall method involves a fair number of moving parts, and it would be good to reconcile them as a single algorithm or a list of bullet points for easy digestion for the reader.
>
> We thank the reviewer for the suggestions and will include the following snippets of pseudo-code in the paper.
> ```python
> def train(x, y, hessian, prior_prec):
> 	# x is the input data
> 	# y is the target data
> 	# parameters is the network parameters
> 	# hessian is hessian of the network
> 	# prior_prec is prior precision. We set it to 1
>
> 	mu_q = parameters
> 	sigma_q = 1 / (hessian + prior_prec)
> 	network_samples = sample_from_normal(mu_q, sigma_q)
> 	for sample in network_samples:
> 		emb = model(x, sample)
> 		pairs = miner(y)
> 		loss += contrastive_loss(emb, pairs)
> 		hessian_batch += hessian_calculator(x, pairs)
>
> 	loss =/ len(network_samples)
> 	hessian_batch =/ len(network_samples)
> 	hessian = alpha * hessian + hessian_batch
> ```
> ```python
> def inference(x, parameters, hessian):
> 	# x is the input data
> 	# parameters is the network parameters
> 	# hessian is hessian of the network
> 	# prior_prec is prior precision. We set it to 1
>
> 	mu_q = parameters
> 	sigma_q = 1 / (hessian + prior_prec)
> 	network_samples = sample_from_normal(mu_q, sigma_q)
> 	emb = []
> 	for sample in network_samples:
> 		emb.append(model(x, sample))
> 	mu,  sigma = vmf_from_samples(emb)
> ```
> We hope this will improve the clarity of the presentation.
>
> > The method can be computationally expensive, and the authors have to resort to construct stochasticity of the last-layer parameters only. Much of the post-hoc LA literature seems to be working with a similar setup, so I do not count this as a major weakness of this paper but rather the whole community.
>
> Bayesian deep learning builds on approximations: We agree with the reviewer that the Bayesian literature tries to approximate the posterior in various ways to come up with computationally feasible and useful methods. We follow common practices in LA and rely on a last-layer, diagonal assumption.
>
> >Minor
> > Please use \citet instead of \citep for references directly referring to the paper, for instance in Line 164.
> > It would be great to have Figure 3 on the same page as the text on ensuring positive definiteness on Page 4.
>
> We thank the reviewer for the suggestions to improve clarity and have updated the paper accordingly.
>
> >Questions:
> > In Line 46, the authors claim that they do not assume any distribution on the stochastic embeddings. But the choice of using Laplacian approximation on the parameters to construct the posterior implicitly makes an assumption on the constructed embeddings. Could the authors clarify this?
>
> The reviewer is right that the Laplace approximation implicitly puts some assumptions on the embedding distribution. Note that these assumptions depend on the choice of the neural network architecture. So the proper statement would be “the Laplace approximation, when tied with a specific neural network architecture, implicitly puts some assumptions on the embedding distribution”. This is exactly what we observe in our case by only considering architecture with a normalization layer at the end: such architecture choice resulted in the implicit constraint of embedding distribution supported on the sphere.
>
> Importantly, no assumptions are present before conditioning on the network architecture. Our text was meant to contrast with existing variational methods that make the Gaussian assumption over the embeddings. Our model makes no such explicit choice, and different instances of the method will result in different assumptions. In principle, with a sufficiently flexible network architecture, it should be possible to obtain any embedding distribution from the Laplace approximation (in a similar spirit to change-of-variables in normalizing flows). In practice, we do observe non-Gaussian unimodal embedding distributions.
>
> > As stated in Eq. (2), the Laplace approximation comes from a second-order Taylor expansion, where the first-order gradient vanishes due to being the optima. Does this does really happen in practice, and how numerically close are we to zero for instance w.r.t. the size of models/data?
>
> The reviewer is right that the gradient is usually ignored in the Laplace approximation, and the weight posterior is sampled from N(mu, H^{-1}). We experimentally find that if we do not ignore the gradient term, and instead sample from N(mu + grad * H^{-1}, H^{-1}), then we get similar results:
>
> |   |  map@1  |  map@5 |  map@10  |  auroc  | auprc | ausc |
> |---|---|---|---|---|---|---|
> | N(mu, H^{-1})  | 0.46 | 0.72 | 0.70 | 0.99 | 1.00 | 0.50 |
> | N(mu + grad * H^{-1}, H^{-1})  | 0.46 | 0.72 | 0.70 | 0.99 | 1.00 | 0.50 |
>
>
> Here, grad is the average gradient over the dataset estimated from one epoch. These results are obtained from LFW, similar to the rest of the ablation studies. We will include it in the ablation table. To answer, how numerically close the gradients are to zero, we provide the min and max of the average gradient: min = -0.0008, max = 0.0008. We believe that the Table and the absolute values of the gradients suggest that it is reasonable to ignore the first-order term.
>
> > Could the authors confirm if my understanding of the overall approach is correct? (a) Using online LA to construct the posterior (b) Use samples from the posterior to construct stochastic embeddings (c) Use embedding samples to construct for the von Mises-Fisher distribution which is used as a single number to quantify uncertainty.
>
> Your understanding is correct. However, note that the last step [fitting the vMF distribution] is optional, e.g. maybe you do not need a single measure of uncertainty but would prefer to work with the samples directly.

---

> > ### Comment · Reviewer_Z8Se · 2023-08-14
> >
> > Thank you for the clarifications. I maintain my accept score.

---

### Official Review · Reviewer_SAQp · 2023-07-10

**Soundness:** 3 good
**Presentation:** 4 excellent
**Contribution:** 3 good
**Rating:** 5
**Confidence:** 4

**Summary:**

They propose a Bayesian encoder for metric learning. They learn a distribution over the network weights with the Laplace approximation. They first prove that the contrastive loss is a negative log-likelihood on the spherical space. They propose three methods that ensure a positive definitive covariance matrix. They present a novel decomposition of the Generalized Gauss-Newton approximation.
The empirical results leads previous methods on OOD examples.

**Strengths:**

- Well organized paper and good writing. Clearly presenting the idea. Code released.
- The bayesian approach for uncertainty measurement in metric learning is interesting.
- The method show improved results, especially on OOD examples, and ablation study is comprehensive.

**Weaknesses:**

- First the experimental results only achieved limited improvement.
- As the paper claimed, The method is slow in computation as it is a bayesian method.
- The paper only focus on a classical contrastive loss which is a margin-based loss, how about other cases? for example, the proxy-based losses?
- Overall the performance improvement is not significant, and the computational efficiency is not competitive, compared with other uncertainty method.

**Questions:**

1. Could you provide a systematic analysis of the computational efficiency?
2. Proxy-anchor loss can be a good case to study with this uncertainty measurement.

**Limitations:**

yes, they have addressed the limitations of the paper.

---

> ### Author Rebuttal · Authors · 2023-08-09
>
> >  First the experimental results only achieved limited improvement.
>
> **Strong UQ performance:** The reviewer is correct that the predictive performance does only improve slightly upon the baselines. However, we highlight that the OOD performance (the focus of the paper) across all 4 datasets improves significantly, e.g. for LFW, Deep Ensemble, PFE, MC dropout has 0.52, 0.03, 0.03 AUROC (recall 0.5 is a random baseline), whereas LAM (post-hoc) and LAM (online) yield 0.65 and 0.71 AUROC. Across all datasets, similar, large performance improvements are observed for uncertainty metrics on OOD and ID.
>
> >  As the paper claimed, The method is slow in computation as it is a bayesian method.
>
> **Computational overhead:** It is true that our approach comes with a higher computational overhead than a standard feedforward network, but it also produces more information: a useful estimate of uncertainty. Depending on the application, this trade-off can be very worth making.
>
> At inference time, we require N forward passes, thus requiring N times more compute than deterministic methods. In practice, however, these N forward passes can easily be parallelized, such that no time overhead might be observed.
>
> >  The paper only focus on a classical contrastive loss which is a margin-based loss, how about other cases? for example, the proxy-based losses?
>
> **Extension to other losses:** The focus of the paper is the contrastive loss, a very common loss in metric learning, which has been shown to perform on par with newer, more sophisticated losses [Metric Learning a Reality Check, Musgrave et al.]. We believe that the method is applicable to the proxy-anchor loss, and this it is an interesting direction to explore in future work.
>
> > Overall the performance improvement is not significant, and the computational efficiency is not competitive, compared with other uncertainty method.
>
> We do not agree. See [Strong UQ performance] above.
>
> >  Questions:
> >  Could you provide a systematic analysis of the computational efficiency?
>
> [see above]
>
> > Proxy-anchor loss can be a good case to study with this uncertainty measurement.
>
> [see above]

---

> > ### Comment · Reviewer_SAQp · 2023-08-19
> >
> > Thanks for providing the rebuttal, which addresses some of the previous concerns. After reading the other reviewers' comments, I slightly improved my rating to Borderline Accept.

---

### Author Rebuttal · Authors · 2023-08-09

We thank the reviewers for their positive and constructive feedback. The reviewers found the problem considered “interesting” [R1] and stated that it is an “important topic for improving the robustness and mitigating the silent failure of deep neural network systems.” [R3] The paper is “well organized” [R1] and with “good writing. Clearly presenting the idea” [R1] “effectively uses figures to demonstrate ideas” [R3], providing “several analogies for various concepts introduced which makes the reader comfortable, and would be a resourceful paper for the community.” [R2] “The use of Laplace approximation [...] in the context of metric learning is novel. This includes the introduction of a probabilistic view of the contrastive loss and the utilization of the Hessian approximation based on GCN.” [R3] They found that the method is “conceptually simple”[R2] yielding “improved results, especially on OOD examples.” [R1] The experimental section provides “Extensive empirical evaluations, including a careful ablation study, are conducted.” [R3] “Most design choices in the method are covered by ablations” [R2] “and ablation study is comprehensive” [R1]. “The results demonstrate that the proposed approach outperforms HIE and PFE in the considered cases.” [R3] We are excited of the reviewers positive reception of the paper, and will address their questions below.

---

### Decision · Program_Chairs · 2023-09-21

**Decision:**

Accept (poster)

**Comment:**

The paper proposes a Bayesian metric learning approach for image retrieval that uses the Laplace approximation to learn a distribution over the weights. The reviewers remarked on the novelty of using a loss based on contrastive learning. They appreciated the empirical evaluation and ablation study, which gives confidence in the improvement made. In general, the idea was broad enough to be applicable to many problems and the reviewers found it likely to appeal to the ML community. Therefore, all reviewers voted to accept the paper.